# Unregularized Linear Convergence in Zero-Sum Game from Preference Feedback

## Abstract

Aligning large language models (LLMs) with human preferences has proven effective for enhancing model capabilities, yet standard preference modeling using the Bradley-Terry model assumes transitivity, overlooking the inherent complexity of human population preferences. Nash learning from human feedback (NLHF) addresses this by framing non-transitive preferences as a two-player zero-sum game, where alignment reduces to finding the Nash equilibrium (NE). However, existing algorithms typically rely on regularization, incurring unavoidable bias when computing the duality gap in the original game. In this work, we provide the first convergence guarantee for Optimistic Multiplicative Weights Update (`OMWU`) in NLHF, showing that it achieves last-iterate linear convergence after a burn-in phase whenever an NE with full support exists, with an instance-dependent linear convergence rate to the original NE, measured by duality gaps. Compared to prior results in Wei et al. (2020), we do not require the assumption of NE uniqueness. Our analysis identifies a novel marginal convergence behavior, where the probability of rarely played actions grows exponentially from exponentially small values, enabling exponentially better dependence on instance-dependent constants than prior results. Experiments corroborate the theoretical strengths of `OMWU` in both tabular and neural policy classes, demonstrating its potential for LLM applications.

## 1 Introduction

The emergence of large language models (LLMs) raises the challenge of aligning model outputs with human preferences. Traditional reinforcement learning (RL) methods require explicit reward functions, which are often difficult to obtain directly. The Bradley-Terry (BT) model (Bradley & Terry, 1952) addresses this by assigning a scalar reward $r(x, y)$ to a response $y$ given a prompt $x$, based on collected preference data. Specifically, the probability of preferring $y$ over $y'$ on prompt $x$ is modeled as $\mathcal{P}(y > y' \mid x) = \sigma(r(x, y) - r(x, y'))$, where $\sigma(t) = 1/(1 + e^{-t})$. Building on this framework, Direct Preference Optimization (DPO, Rafailov et al. (2023)) leverages the closed-form solution to fine-tune the policy.

However, the BT model implicitly assumes that human preferences are transitive. Empirical studies (May, 1954) provide evidence of intransitivity at the population level. To address this limitation, Munos et al. (2023) introduced *Nash Learning from Human Feedback* (NLHF), which formulates alignment as finding a Nash equilibrium (NE) of the preference game. Since $\mathcal{P}(y > y' \mid x) + \mathcal{P}(y' > y \mid x) = 1$, the problem naturally reduces to a two-player constant-sum game.

For NLHF to be deployable in the LLM setting, the algorithm must satisfy additional requirements:

First, it must achieve *last-iterate convergence*, meaning the final policy produced by training is close to a Nash equilibrium. In contrast, *average-iterate convergence* guarantees only that the time-averaged policy approximates equilibrium, which is impractical for deployment.

Second, the algorithm should be implementable with neural network parameterizations. This rules out certain methods such as Optimistic Gradient Descent Ascent (OGDA, Wei et al. (2020)), which requires orthogonal projections onto the probability simplex.

Table 1: Comparison of `OMWU` results in the NLHF setting. The comparison between "exponential dependence" and "polynomial dependence" is explained towards end of Section 1.1.

| Assumptions | Unique Equilibrium | Multiple Equilibria Full-Support Equilibrium |
|---|---|---|
| **Full-Support Equilibrium Exists** | Linear convergence exponential dependence (Wei et al., 2020) **polynomial dependence (ours)** | **Linear convergence polynomial dependence (ours)** |

Third, the algorithm should avoid nested optimization, e.g., $\boldsymbol{\theta}^{(t+1)} = \arg\min_{\boldsymbol{\theta}} \mathcal{L}_{\text{inner}}(\boldsymbol{\theta}; \boldsymbol{\theta}^{(t)})$, which is widely adopted by almost all prior works in NLHF (Munos et al., 2023; Swamy et al., 2024; Wu et al., 2024; Shani et al., 2024; Zhang et al., 2024; Wang et al., 2024; Zhang et al., 2025). In practice, there is a trade-off between the number of gradient descent steps on $\mathcal{L}_{\text{inner}}$ to approximate $\widehat{\theta}^{(t+1)}$ and the approximation error, while more steps incur significantly higher computational overhead.

Motivated by these requirements, we revisit the *Optimistic Multiplicative Weights Update* (`OMWU`, Daskalakis & Panageas (2018)) algorithm to assess its potential for more complex tasks such as NLHF. Wei et al. (2020) proved that under the assumption of a unique NE, `OMWU` achieves last-iterate linear convergence after a burn-in period, with an instance-dependent convergence rate for two-player zero-sum games. However, general preference matrices typically have infinitely many NEs. Furthermore, their analysis yields burn-in times and convergence rates with *exponential* dependence on instance-dependent constants, which may be prohibitive for NLHF tasks.

## 1.1 MAIN CONTRIBUTIONS

This paper provides new theoretical guarantees for `OMWU` in the NLHF setting. Our contributions are as follows:

• **Improved efficiency under milder assumptions.** Under the natural assumption that a full-support Nash equilibrium exists (rather than the uniqueness of NE), we establish polynomial (rather than exponential) dependence on instance-dependent constants for both convergence rate and burn-in time, while maintaining last-iterate linear convergence over the number of updates. This substantially reduces concerns about the practicality of `OMWU` for NLHF.

• **New analytical framework for escaping behavior.** We introduce a novel method for analyzing how `OMWU` escapes undesirable regions of the strategy space. To our knowledge, this is the first work to formalize such escaping dynamics, even beyond `OMWU`, opening a new direction for understanding dynamics in game-theoretical problems.

We compare our results with those of Wei et al. (2020) in Table 1.

Under the unique equilibrium assumption, the orders of the burn-in time and convergence rate in Wei et al. (2020) are

$$O\left(\frac{\ln(A)}{\eta^4 C_{\boldsymbol{P}}^2 \exp(-4\ln(A)/\varepsilon)}\right) \text{ and } O\left(\eta^2 C_{\boldsymbol{P}}^2 \exp(-3\ln(A)/\varepsilon)\right)$$

when initialization is uniform, which is in stark contrast with our orders

$$O\left(\frac{D_{\text{KL}}(\boldsymbol{\pi}^*\|\hat{\boldsymbol{\pi}}^{(1)})^3(\eta L)^2 A^2}{(1 - 4\eta^2 L^2)C_{\boldsymbol{P}}^4 \varepsilon^6} \cdot \max\left\{1, \frac{C_{\boldsymbol{P}}^2 \varepsilon^4 A}{(\eta L)^2}\right\}^3\right) \text{ and } O(\eta^2 \varepsilon^3 C_{\boldsymbol{P}}^2),$$

which does not depend exponentially on $\ln(A)/\varepsilon$. Here, $A$ is the size of action space, $\hat{\boldsymbol{\pi}}^{(1)}$ is the initialization, $\boldsymbol{\pi}^*$ is the equilibrium, $\eta$ is the learning rate, and readers may reder to Section 3 for the precise definitions of $\varepsilon$, $L$, and $C_{\boldsymbol{P}}$.

## 1.2 PAPER OVERVIEW

We begin by introducing the NLHF framework and the `OMWU` algorithm in Section 3. In Section 4, we present our main theoretical result, together with an overview of the convergence behavior and the analytical techniques used in our proofs. We then validate our theory with empirical results in Section 5.

Table 2: Comparison of algorithms with convergence guarantees to $\delta$-NE in NLHF. **Convergence to $\delta$-NE:** the number of steps required to find a policy with duality gap at most $\delta$. $\tilde{O}$ hides $\text{poly}(\log(|\mathbb{A}|/\eta))$ factors. When $\text{poly}(\delta^{-1})$ terms exist, $\text{poly}(\log(\delta^{-1}))$ factors are also hidden. **Last-iterate convergence:** "Yes" indicates that the convergence rate applies to the final policy; "No" indicates that it applies only to the average policy. **Efficient update:** "Yes" indicates a single-step gradient update suffices, while "No" indicates nested optimization is required.

| Algorithm | Convergence to $\delta$-NE | Last-iterate Convergence | Efficient Update |
|:---:|:---:|:---:|:---:|
| OMD | $\tilde{O}(\delta^{-2})$ | No | **Yes** |
| SPPO (Wu et al., 2024) | $\tilde{O}(\delta^{-2})$ | No | No |
| MPO (Wang et al., 2024) | $\tilde{O}(\delta^{-2})$ | **Yes** | No |
| ONPO (Zhang et al., 2025) | $\tilde{O}(\delta^{-1})$ | No | No |
| EGPO (Zhou et al., 2025) | $\tilde{O}(\delta^{-1})$ | **Yes** | **Yes** |
| OMWU | $\tilde{O}(\log(\delta^{-1}))$ **(linear)** | **Yes** | **Yes** |

## 2 RELATED WORKS

**NLHF.** Since the introduction of the NLHF framework by Munos et al. (2023), a growing body of work has studied algorithms for solving NLHF, most of which rely on regularization. The Nash-MD algorithm proposed in Munos et al. (2023) achieves linear convergence in the regularized setting. The MPO algorithm Wang et al. (2024) provides an $\tilde{O}(\delta^{-2})$ last-iterate convergence to a $\delta$-NE. More recently, the EGPO algorithm of Zhou et al. (2025) achieves an improved $\tilde{O}(\delta^{-1})$ convergence rate compared to MPO, while eliminating the need for nested optimization. Additional studies of NLHF include Wu et al. (2024); Swamy et al. (2024); Ye et al. (2024); Rosset et al. (2024); Calandriello et al. (2024); Zhang et al. (2024; 2025). Table 2 summarizes the theoretical guarantees of several algorithms in the NLHF setting.

**Computing Nash equilibria in two-player zero-sum games.** Computing Nash equilibria in two-player zero-sum games was a central research topic long before NLHF was proposed. Online Mirror Descent (OMD, Cesa-Bianchi & Lugosi (2006); Lattimore & Szepesvári (2020)), originally developed for online convex learning, naturally applies to this setting but guarantees only average-iterate convergence. In contrast, OMWU and Optimistic Gradient Descent Ascent (OGDA) have been shown to enjoy instance-dependent linear convergence (Wei et al., 2020). Among these methods, OGDA uses direct parametrization (using $\theta_{x,y}$ directly as the probability), making it less applicable; however, its convergence holds even when the equilibrium is non-unique.

**OMWU.** The Optimistic Multiplicative Weights Update (OMWU) algorithm was first proposed by Daskalakis & Panageas (2018), who established its last-iterate convergence under the uniqueness assumption, though without a convergence rate. Later, Wei et al. (2020) proved linear convergence at an instance-dependent rate for general saddle-point optimization problems under the same assumption. On the negative side, Cai et al. (2024) showed that last-iterate convergence must be arbitrarily slow for a broad class of algorithms including OMWU, even in simple two-action settings. Other studies of OMWU include Lee et al. (2021); Daskalakis et al. (2021); Anagnostides et al. (2022).

## 3 PRELIMINARIES

### 3.1 MULTI-ARMED BANDITS

A multi-armed bandit has an action space $\mathbb{A}$ (the reward function is irrelevant in our setting). A contextual bandit additionally has a context space $\mathbb{X}$, where the agent chooses an action $a \in \mathbb{A}$ for each context $x \in \mathbb{X}$. In our fine-tuning setting, $\mathbb{X}$ corresponds to the prompt space and $\mathbb{A}$ to the response space. For clarity, we state our theory in the multi-armed bandit setting.

A policy $\boldsymbol{\pi}$ is a probability distribution over $\mathbb{A}$. For computational convenience, we parametrize $\boldsymbol{\pi}$ with a vector $\boldsymbol{\theta}$ such that

$$\pi_a = \frac{\exp(\theta_a)}{\sum_{a' \in \mathbb{A}} \exp(\theta_{a'})}.$$

Note that $\boldsymbol{\theta}$ is a valid parametrization of $\boldsymbol{\pi}$ if and only if it differs from $\log \boldsymbol{\pi}$ by a constant shift.

### 3.2 NLHF

In the NLHF problem (see Munos et al. (2023)), each pair of actions $a, a' \in \mathbb{A}$ is associated with a probability $\mathcal{P}(a > a')$, representing the probability that $a$ is preferred over $a'$. Clearly, $\mathcal{P}(a > a') + \mathcal{P}(a' > a) = 1$ when $a \neq a'$, and we set $\mathcal{P}(a > a) = \frac{1}{2}$ so that this relation also holds for $a = a'$.

We define the preference matrix $\boldsymbol{P}$ as

$$P_{a,a'} = \mathcal{P}(a > a') - \tfrac{1}{2}.$$

By construction, $\boldsymbol{P}$ is skew-symmetric $(\boldsymbol{P} + \boldsymbol{P}^\top = \boldsymbol{0})$[1].

The goal of NLHF is to find a policy $\boldsymbol{\pi}^*$ that maximizes its probability of being preferred against an adversarial policy $\boldsymbol{\pi}'$, i.e.,

$$\boldsymbol{\pi}^* = \arg\max_{\boldsymbol{\pi}} \min_{\boldsymbol{\pi}'} P(\boldsymbol{\pi} > \boldsymbol{\pi}'),$$

where

$$P(\boldsymbol{\pi} > \boldsymbol{\pi}') = \mathbb{E}_{a \sim \boldsymbol{\pi}, a' \sim \boldsymbol{\pi}'} P(a > a') = \boldsymbol{\pi}^\top \boldsymbol{P} \boldsymbol{\pi}' + \tfrac{1}{2}.$$

By von Neumann's minimax theorem v. Neumann (1928),

$$\max_{\boldsymbol{\pi}} \min_{\boldsymbol{\pi}'} \boldsymbol{\pi}^\top \boldsymbol{P} \boldsymbol{\pi}' = \min_{\boldsymbol{\pi}'} \max_{\boldsymbol{\pi}} \boldsymbol{\pi}^\top \boldsymbol{P} \boldsymbol{\pi}',$$

and since $\boldsymbol{P}$ is skew-symmetric,

$$\max_{\boldsymbol{\pi}} \min_{\boldsymbol{\pi}'} \boldsymbol{\pi}^\top \boldsymbol{P} \boldsymbol{\pi}' = \min_{\boldsymbol{\pi}'} \max_{\boldsymbol{\pi}} \boldsymbol{\pi}^\top \boldsymbol{P} \boldsymbol{\pi}' = 0 = \boldsymbol{a}^\top \boldsymbol{P} \boldsymbol{a} \quad \forall \boldsymbol{a}.$$

Thus, if $\boldsymbol{\pi}^*$ is an optimizer of the minimax objective, then

$$\min_{\boldsymbol{\pi}'} (\boldsymbol{\pi}^*)^\top \boldsymbol{P} \boldsymbol{\pi}' = \max_{\boldsymbol{\pi}} \boldsymbol{\pi}^\top \boldsymbol{P} \boldsymbol{\pi}^* = 0.$$

In this case, we call $\boldsymbol{\pi}^*$ a Nash equilibrium of $\boldsymbol{P}$[2].

We denote the set of all Nash equilibria by $\mathbb{M}$. For any policy $\boldsymbol{\pi}$, the duality gap is defined as

$$\mathsf{DualGap}(\boldsymbol{\pi}) = \max_{\boldsymbol{\pi}'} (\boldsymbol{\pi}')^\top \boldsymbol{P} \boldsymbol{\pi} - \min_{\boldsymbol{\pi}'} \boldsymbol{\pi}^\top \boldsymbol{P} \boldsymbol{\pi}' = 2 \max_{a \in \mathbb{A}} (\boldsymbol{P}\boldsymbol{\pi})_a.$$

The duality gap is always nonnegative and equals zero if and only if $\boldsymbol{\pi}$ is a Nash equilibrium. We say a policy $\boldsymbol{\pi}$ is $\delta$-NE when $\mathsf{DualGap}(\boldsymbol{\pi}) \leqslant \delta$.

### 3.3 Optimistic multiplicative weights update (OMWU)

The Optimistic Multiplicative Weights Update (OMWU) algorithm was introduced by Daskalakis & Panageas (2018) for solving equilibria in zero-sum games and later extended to saddle-point problems. In the NLHF setting, the algorithm simplifies to

$$\boldsymbol{\pi}^{(t)} = \arg\min_{\boldsymbol{\pi}} \left\{ \eta \langle \boldsymbol{\pi}, \boldsymbol{P}\boldsymbol{\pi}^{(t-1)} \rangle + D_{\mathrm{KL}}(\boldsymbol{\pi} \| \hat{\boldsymbol{\pi}}^{(t)}) \right\},$$

---

[1] Some works define the preference matrix without subtracting $\frac{1}{2}$. This choice has little impact in practice, but in our case, the current definition ensures skew-symmetry.

[2] Formally, a Nash equilibrium of a zero-sum game matrix $\boldsymbol{P}$ is a pair $(\boldsymbol{\pi}_1, \boldsymbol{\pi}_2)$ such that $\boldsymbol{\pi}_1^\top \boldsymbol{P} \boldsymbol{\pi}_2 = \min_{\boldsymbol{\pi}'} \boldsymbol{\pi}_1^\top \boldsymbol{P} \boldsymbol{\pi}' = \max_{\boldsymbol{\pi}'} (\boldsymbol{\pi}')^\top \boldsymbol{P} \boldsymbol{\pi}_2$. For skew-symmetric $\boldsymbol{P}$, $(\boldsymbol{\pi}_1, \boldsymbol{\pi}_2)$ is a Nash equilibrium if and only if $(\boldsymbol{\pi}_1, \boldsymbol{\pi}_1)$ and $(\boldsymbol{\pi}_2, \boldsymbol{\pi}_2)$ are Nash equilibria. Hence, our notational simplification is justified.

$$\hat{\boldsymbol{\pi}}^{(t+1)} = \arg\min_{\boldsymbol{\pi}} \left\{ \eta\langle\boldsymbol{\pi}, \boldsymbol{P}\boldsymbol{\pi}^{(t)}\rangle + D_{\mathrm{KL}}(\boldsymbol{\pi}\|\hat{\boldsymbol{\pi}}^{(t)}) \right\},$$

where $\eta$ is the learning rate, and $\boldsymbol{\pi}^{(0)} = \hat{\boldsymbol{\pi}}^{(1)}$ are initializations.

In the parametrized form, OMWU reduces to

$$\boldsymbol{\theta}^{(t)} = \boldsymbol{\theta}^{(t-1)} + \eta\boldsymbol{P}\hat{\boldsymbol{\pi}}^{(t)}, \tag{1}$$

$$\hat{\boldsymbol{\theta}}^{(t+1)} = \boldsymbol{\theta}^{(t)} + \eta\boldsymbol{P}\hat{\boldsymbol{\pi}}^{(t)}. \tag{2}$$

### 3.4 REGULARIZATION IN NLHF

Although regularization is not part of OMWU, it appears in related algorithms. For a reference policy $\boldsymbol{\pi}_{\mathrm{ref}}$, define

$$\mathsf{Dualgap}_\beta(\boldsymbol{\pi}) = \max_{\boldsymbol{\pi}'}\left((\boldsymbol{\pi}')^\top\boldsymbol{P}\boldsymbol{\pi} - \beta D_{\mathrm{KL}}(\boldsymbol{\pi}'\|\boldsymbol{\pi}_{\mathrm{ref}}) + \beta D_{\mathrm{KL}}(\boldsymbol{\pi}\|\boldsymbol{\pi}_{\mathrm{ref}})\right)$$
$$- \min_{\boldsymbol{\pi}''}\left(\boldsymbol{\pi}^\top\boldsymbol{P}\boldsymbol{\pi}'' - \beta D_{\mathrm{KL}}(\boldsymbol{\pi}\|\boldsymbol{\pi}_{\mathrm{ref}}) + \beta D_{\mathrm{KL}}(\boldsymbol{\pi}''\|\boldsymbol{\pi}_{\mathrm{ref}})\right).$$

This transforms NLHF into a convex optimization problem in $\mathsf{Dualgap}_\beta(\boldsymbol{\pi})$ at the cost of introducing regularization error.

### 3.5 ASSUMPTIONS

Throughout, we impose the following assumption on $\boldsymbol{P}$:

**Assumption 1.** *For every $a \in \mathbb{A}$, there exists a Nash equilibrium $\boldsymbol{\pi}$ of $\boldsymbol{P}$ such that $\pi_a > 0$.*

**Remark 1.** This is equivalent to the existence of a Nash equilibrium $\boldsymbol{\pi}$ with $\pi_a > 0$ for all $a \in \mathbb{A}$. It is well known that the set $\mathbb{M}$ of Nash equilibria is convex.

We define the KL projection of a policy $\boldsymbol{\pi}$ onto $\mathbb{M}$ as

$$p(\boldsymbol{\pi}) = \arg\min_{\boldsymbol{\pi}'\in\mathbb{M}} D_{\mathrm{KL}}(\boldsymbol{\pi}'\|\boldsymbol{\pi}).$$

Under Assumption 1, we obtain the following:

**Lemma 1.** *If $\boldsymbol{\pi}$ satisfies $\pi_a > 0$ for all $a \in \mathbb{A}$, then $p(\boldsymbol{\pi})$ is well-defined, unique, and satisfies $p(\boldsymbol{\pi})_a > 0$ for all $a$.*

**Lemma 2.** *Under Assumption 1, the OMWU algorithm satisfies*

$$p(\hat{\boldsymbol{\pi}}^{(1)}) = p(\hat{\boldsymbol{\pi}}^{(2)}) = \cdots = p(\hat{\boldsymbol{\pi}}^{(t)}) = \cdots.$$

Proofs are deferred to Appendix A. By Lemma 2, we define

$$\boldsymbol{\pi}^* = p(\hat{\boldsymbol{\pi}}^{(t)}).$$

**Remark 2.** If Lemma 2 holds and the sequence $\hat{\boldsymbol{\pi}}^{(t)}$ converges as $t \to \infty$, then the limit must be $\boldsymbol{\pi}^*$. This plays the role of the uniqueness assumption in prior work, allowing us to predict the convergence point without taking infinite steps. Moreover, with uniform initialization, $\boldsymbol{\pi}^*$ corresponds to the equilibrium with minimal negative entropy.

Finally, we introduce constants used in later proofs:

**Definition 1** (Instance-dependent constants).

$$\varepsilon = \min_{a\in\mathbb{A}}\pi_a^*, \quad L = \max_{a,a'\in\mathbb{A}}|P_{a,a'}|, \quad C_{\boldsymbol{P}} = \min_{\boldsymbol{\pi}\in\Delta(\mathbb{A})\setminus\mathbb{M}}\frac{\|\boldsymbol{P}\boldsymbol{\pi}\|_\infty}{\|\boldsymbol{\pi} - p(\boldsymbol{\pi})\|_1}.$$

Clearly $\varepsilon > 0$ by Assumption 1, and the proof of $C_{\boldsymbol{P}} > 0$ is given in Appendix B.

The constant $C_{\boldsymbol{P}}$ is novel but crucial. This conveys the idea that if some policy $\boldsymbol{\pi}$ is far from the predicted convergence point $p(\boldsymbol{\pi})$, then there must be a large update following OMWU algorithm in the parametrized space at some coordinate.

## 4 MAIN THEOREM AND PROOF SKETCH

### 4.1 STATEMENT OF MAIN THEOREM

Our main contribution is to establish a last-iterate linear convergence guarantee for OMWU in zero-sum games with a full-support Nash equilibrium. This extends the uniqueness-based result of Wei et al. (2020), providing a broader characterization of OMWU dynamics.

**Theorem 2.** *For any skew-symmetric matrix $\boldsymbol{P}$ satisfying Assumption 1, any initialization $\hat{\boldsymbol{\pi}}^{(1)}$, and any learning rate $\eta$ such that $\eta L < \frac{1}{2}$, there exists a burn-in time*

$$
T = O\left( \frac{D_{\mathrm{KL}}(\boldsymbol{\pi}^*\|\hat{\boldsymbol{\pi}}^{(1)})^3 (\eta L)^2 |\mathbb{A}|^2}{(1 - 4\eta^2 L^2) C_{\boldsymbol{P}}^4 \varepsilon^6} \cdot \max\left\{ 1, \frac{C_{\boldsymbol{P}}^2 \varepsilon^4 |\mathbb{A}|}{(\eta L)^2} \right\}^3 \right)
$$

*such that*

$$
D_{\mathrm{KL}}(\boldsymbol{\pi}^*\|\hat{\boldsymbol{\pi}}^{(t)}) \leqslant \varepsilon \exp\left( -O\left( \frac{\eta^2 \varepsilon^3 C_{\boldsymbol{P}}^2 (t - T)}{(1 - 4\eta^2 L^2)} \right) \right)
$$

*for all $t \geqslant T$.*

A complete proof is given in Appendix C. Here we highlight the key ideas behind the analysis.

**Remark 3.** Several observations follow from our theorem:

- Although the result is expressed in terms of $D_{\mathrm{KL}}(\boldsymbol{\pi}^*\|\hat{\boldsymbol{\pi}}^{(t)})$, the guarantee directly implies convergence in terms of the duality gap via Pinsker's inequality (Lemma 7):

$$
D_{\mathrm{KL}}(\boldsymbol{\pi}^*\|\hat{\boldsymbol{\pi}}^{(t)}) \geqslant \frac{1}{4L} \mathsf{DualGap}(\hat{\boldsymbol{\pi}}^{(t)})^2.
$$

- The burn-in time $T$ simplifies under mild assumptions. For uniform initialization, $D_{\mathrm{KL}}(\boldsymbol{\pi}^*\|\hat{\boldsymbol{\pi}}^{(1)}) = O(\log |\mathbb{A}|)$. Moreover, since $\varepsilon < |\mathbb{A}|$ under Assumption 1 and $C_{\boldsymbol{P}} \leqslant 1$, the term $\frac{C_{\boldsymbol{P}}^2 \varepsilon^4 |\mathbb{A}|}{(\eta L)^2}$ is typically smaller than 1, except when $\eta$ is chosen extremely small.

### 4.2 PROOF SKETCH

**Non-increasing KL-projection.** We begin by showing that the quantity $D_{\mathrm{KL}}(\boldsymbol{\pi}^*\|\hat{\pi}_t)$ decreases over time, up to a small perturbation. Specifically, define

$$
\Theta_t = D_{\mathrm{KL}}(\boldsymbol{\pi}^*\|\hat{\boldsymbol{\pi}}^{(t)}) + 4\eta^2 L^2 D_{\mathrm{KL}}(\hat{\boldsymbol{\pi}}^{(t)}\|\boldsymbol{\pi}^{t-1}).
$$

It can be shown that

$$
\Theta_t - \Theta_{t+1} \geqslant (1 - 4\eta^2 L^2)\left( D_{\mathrm{KL}}(\boldsymbol{\pi}^{(t)}\|\hat{\boldsymbol{\pi}}^{(t)}) + D_{\mathrm{KL}}(\hat{\boldsymbol{\pi}}^{(t+1)}\|\boldsymbol{\pi}^{(t)}) \right).
$$

This inequality, adapted from Lemma 10 of Wei et al. (2020), ensures that $\Theta_t$ strictly decreases whenever $\eta < 1/(2L)$. The main task is therefore to quantify how fast $\Theta_t$ converges to 0.

Figure 1 illustrates the trajectory of $\Theta_t - \Theta_{t+1}$ on a cyclic-preference matrix with initialization near $(1, 0, 0)$. The dynamics naturally split into two phases:

- a *burn-in stage*, with oscillatory behavior;
- a *convergence stage*, with nearly linear decay.

**Burn-in Stage: Subgame Case.** The subgame case arises when some action $a$ satisfies both: (i) $\hat{\pi}_a^{(t)}$ (or $\hat{\pi}_a^{(t+1)}$) is large, and (ii) the update $|\eta(\boldsymbol{P}\boldsymbol{\pi}^{(t)})_a|$ is also large. Here the policy shifts significantly. Define

$$
K^{(t+1)} = \max_{a \in \mathbb{A}} \hat{\pi}_a^{(t+1)} |\eta(\boldsymbol{P}\boldsymbol{\pi}^{(t)})_a|, \quad \hat{K}^{(t+1)} = \max_{a \in \mathbb{A}} \hat{\pi}_a^{(t)} |\eta(\boldsymbol{P}\boldsymbol{\pi}^{(t)})_a|.
$$

One can then show that there exists a constant $C' > 0$ such that

$$
D_{\mathrm{KL}}(\boldsymbol{\pi}^{(t)}\|\hat{\boldsymbol{\pi}}^{(t)}) + D_{\mathrm{KL}}(\hat{\boldsymbol{\pi}}^{(t+1)}\|\boldsymbol{\pi}^{(t)}) \geqslant C' \max\{\hat{K}^{(t+1)}, K^{(t+1)}\}^2,
$$

following the argument of Appendix D.3 in Wei et al. (2020).

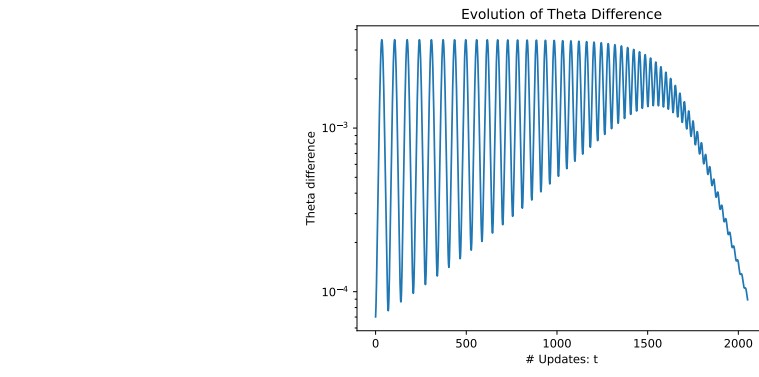

Figure 1: Evolution of $\Theta_t - \Theta_{t+1}$ for $\boldsymbol{P} = \begin{pmatrix} 0 & 1/2 & -1/2 \\ -1/2 & 0 & 1/2 \\ 1/2 & -1/2 & 0 \end{pmatrix}$ with initialization near $(1, 0, 0)$.

**Burn-in Stage: Marginal Case.** In contrast, the marginal case occurs when neither $K^{(t+1)}$ nor $\hat{K}^{(t+1)}$ is large, so updates in probability space are small. However, if $D_{\mathrm{KL}}(\boldsymbol{\pi}^* \| \hat{\boldsymbol{\pi}}^{(t)})$ is still large, at least one coordinate of $\hat{\boldsymbol{\pi}}^{(t)}$ must drift in its *logarithmic* value, ensuring continued progress. To capture this, we introduce the potential

$$\Phi_t = \frac{\langle \log \hat{\boldsymbol{\pi}}^{(t)} - \log \boldsymbol{\pi}^*, \eta \boldsymbol{P} \hat{\boldsymbol{\pi}}^{(t)} \rangle}{(\Theta_t + 2\mathrm{e}^{-1})^2}$$

and analyze $\Phi_t - \Phi_{t+1}$. The leading term reduces to

$$\langle \eta \boldsymbol{P} \boldsymbol{\pi}^{(t)}, \eta \boldsymbol{P} \boldsymbol{\pi}^{(t)} \rangle \geqslant \| \eta \boldsymbol{P} \boldsymbol{\pi}^{(t)} \|_\infty^2,$$

using the update rule and the closeness of $\boldsymbol{\pi}^{(t)}$ and $\hat{\boldsymbol{\pi}}^{(t+1)}$ (guaranteed by small $\hat{K}^{(t+1)}$). After controlling approximation and residual terms, we obtain

$$\Phi_t - \Phi_{t+1} \geqslant \frac{\| \eta \boldsymbol{P} \boldsymbol{\pi}^{(t)} \|_\infty^2}{(\Theta_t + 2\mathrm{e}^{-1})^2} - O\left( \frac{\eta L |\mathbb{A}| (K^{(t)} + \hat{K}^{(t+1)} + K^{(t+1)})}{\varepsilon (\Theta_t + 2\mathrm{e}^{-1})^2} \right) - (\text{minor terms}).$$

**Remark 4.** The dependence on $\varepsilon$ arises naturally from Assumption 1. Intuitively, if $\hat{\boldsymbol{\pi}}^{(t)}$ and $\boldsymbol{\pi}^{(t)}$ concentrate near a non-equilibrium policy $\boldsymbol{\pi}'$ with restricted support, then escaping requires roughly

$$\min_{a: (\boldsymbol{P}\boldsymbol{\pi}')_a > 0} \frac{-\log \hat{\boldsymbol{\pi}}_a^{(t)}}{(\boldsymbol{P}\boldsymbol{\pi}')_a}$$

steps. If $\boldsymbol{\pi}_a^* = 0$, then $-\log \hat{\boldsymbol{\pi}}_a^{(t)} \to \infty$, making such bounds impossible without the assumption $\varepsilon > 0$.

**Burn-in Stage: Combined Analysis.** To unify both subcases, we introduce an augmented expression

$$\bar{\Theta}_t = \Theta_{t-1} + c_1 \Theta_t + c_2 \Phi_t,$$

for constants $c_1, c_2 > 0$. One can show

$$\bar{\Theta}_t - \bar{\Theta}_{t+1} \geqslant O\left( \frac{(1 - 4\eta^2 L^2) C_{\boldsymbol{P}}^4 \varepsilon^6}{\eta^2 L^2 |\mathbb{A}|^2 (\Theta_t + 2\mathrm{e}^{-1})^2} \right)$$

whenever $\Theta_t \geqslant \varepsilon$, while ensuring $\bar{\Theta}_t > \Theta_t$. Consequently, we obtain

$$\Theta_T < \varepsilon \quad \text{for} \quad T = O\left( \frac{\bar{\Theta}_1^3 (\eta L)^2 |\mathbb{A}|^2}{(1 - 4\eta^2 L^2) C_{\boldsymbol{P}}^4 \varepsilon^6} \right).$$

**Convergence Stage.** Once $\Theta_T < \varepsilon$, we can use the subgame bound together with lower bounds on $\min_a \hat{\pi}_a^{(t+1)}$ and $\max_a |(\boldsymbol{P}\boldsymbol{\pi}^{(t)})_a|$ to establish exponential convergence:

$$D_{\mathrm{KL}}(\boldsymbol{\pi}^* \| \hat{\boldsymbol{\pi}}^{(t)}) \leqslant \varepsilon \exp\big(-O(\eta^2 \varepsilon^3 C_{\boldsymbol{P}}^2 (t - T))\big).$$

**Remark 5.** Neither stage is analyzed tightly. In the burn-in phase, the lower bound on $\bar{\Theta}_t - \bar{\Theta}_{t+1}$ stems mainly from the transition region, which may not dominate in practice. In the convergence stage, our rate can be loose when $\arg\min_a \hat{\pi}_a^{(t+1)} \neq \arg\max_a |(\boldsymbol{P}\boldsymbol{\pi}^{(t)})_a|$. Improving both bounds is left for future work.

## 5 EXPERIMENTS

We present simulation results comparing the performance of OMWU against several existing NLHF algorithms (OMD, SPPO, MPO, ONPO, and EGPO).

### 5.1 GAME MATRICES

We construct $\mathcal{P}$ as follows: fix $n$ (the matrix size) and an integer $0 \leqslant m < n$ which roughly denotes the rank of the NE polytope. Generate $m$ random policies, and let $\boldsymbol{M}$ be the $n \times (n - m)$ matrix whose columns span their orthogonal complement. Then generate a random $(n-m) \times (n-m)$ skew-symmetric matrix $\boldsymbol{A}$ and set $\boldsymbol{P} = \boldsymbol{M}\boldsymbol{A}\boldsymbol{M}^\top$. Finally, normalize $\boldsymbol{P}$ so that $\boldsymbol{P} + \frac{1}{2}$ is nonnegative. Refer to Algorithm 1 for more details. We choose $n = 10$ for **tabular** and $n = 100$ for **neural** policy setting, respectively, and $0 \leqslant m < n/2$. Note that Assumption 1 can fail when $m = 0$.

### 5.2 IMPLEMENTATION

Using the notation in Zhou et al. (2025), OMWU update can be written as: $\boldsymbol{\theta}^{(-1/2)} = \boldsymbol{\theta}^{(0)} = \mathbf{0}$, for $t \geqslant 0$,

$$\boldsymbol{\theta}^{(t+1/2)} = \boldsymbol{\theta}^{(t)} + \eta \boldsymbol{P} \boldsymbol{\pi}^{(t-1/2)},$$
$$\boldsymbol{\theta}^{(t+1)} = \boldsymbol{\theta}^{(t)} + \eta \boldsymbol{P} \boldsymbol{\pi}^{(t+1/2)}.$$

Define a generalized IPO (Azar et al., 2023) loss using separate distributions for $(y, y')$ and $y''$ (here we assume they are independent of $\theta$, and simply choose $\beta = 1$):

$$\mathcal{L}_{\mathrm{IPO}}(\boldsymbol{\theta}; \boldsymbol{\pi}_{\mathrm{ref}}, \rho, \mu) = \mathbb{E}_{(y,y') \sim \rho}\left[\left(\log \frac{\boldsymbol{\pi}_{\boldsymbol{\theta}}(y)\boldsymbol{\pi}_{\mathrm{ref}}(y')}{\boldsymbol{\pi}_{\boldsymbol{\theta}}(y')\boldsymbol{\pi}_{\mathrm{ref}}(y)} - \mathbb{E}_{y'' \sim \mu}[\mathcal{P}(y > y'') - \mathcal{P}(y' > y'')]\right)^2\right].$$

The result in Section 4.2 of Zhou et al. (2025) gives us the update:

$$\boldsymbol{\theta}^{(t+1/2)} = \boldsymbol{\theta}^{(t)} - \underbrace{\frac{\eta_{\mathrm{theory}} |\mathbb{A}|}{4}}_{=: \eta_{\mathrm{optimizer}}} \nabla_{\boldsymbol{\theta}} \mathcal{L}_{\mathrm{IPO}}(\boldsymbol{\theta}^{(t)}; \textcolor{red}{\mathsf{sg}[\boldsymbol{\pi}^{(t)}]}, \mathsf{Uniform}, \boldsymbol{\pi}^{(t-1/2)}),$$

$$\boldsymbol{\theta}^{(t+1)} = \boldsymbol{\theta}^{(t)} - \frac{\eta_{\mathrm{theory}} |\mathbb{A}|}{4} \nabla_{\boldsymbol{\theta}} \mathcal{L}_{\mathrm{IPO}}(\boldsymbol{\theta}^{(t)}; \textcolor{red}{\mathsf{sg}[\boldsymbol{\pi}^{(t)}]}, \mathsf{Uniform}, \boldsymbol{\pi}^{(t+1/2)}),$$

where Uniform is the uniform distribution over $\mathbb{A} \times \mathbb{A}$, and $\mathsf{sg}[\cdot]$ means stopping-gradient.

The choice of codebase and implementation of the baselines are detailed in Appendix D.2. The hyperparameters when running the experiments are listed in Appendix D.3.

### 5.3 RESULTS

We select one matrix from the tabular and neural policy setting, respectively. They are shown in Figure 2, while the full experiment results are shown in Appendix D.4. In the title of each experiment, we report $\varepsilon$ and $\lambda_{\min}$, the smallest positive singular value of $\mathcal{P}$, which is related to the constant $C_{\boldsymbol{P}}$. Refer to Definition 1 for the definitions of $\varepsilon$ and $C_{\boldsymbol{P}}$.

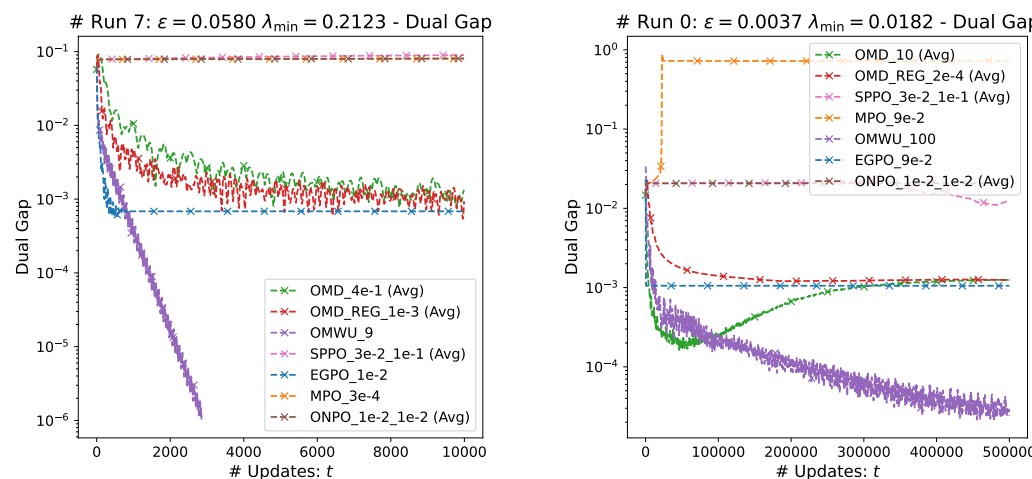

Figure 2: Selected results under the **tabular** (left) and **neural** (right) policy setting.

## 5.4 REMARKS

We make several remarks based on results in this section as well as those in Appendix D.4.

• In tabular setting, `OMWU` consistently achieves **linear** convergence in terms of the duality gap across all matrices, corroborating our main theorem. In neural setting, `OMWU` still outperforms other baselines. Meanwhile, regularized algorithms can only converge to the value related to the regularization coefficient $\beta$. Convergence of `OMWU` accelerates when $\lambda_{\min}$ is large.

• Figure 3 illustrates the stark contrast between the **last-iterate** and **average-iterate** behavior of `OMD` (regularized). This result highlights the advantage of algorithms with last-iterate convergence guarantees.

• In our constructed game matrices, **all** the algorithms requiring **nested-optimization** (`SPPO`, `MPO`, `ONPO`) fail to converge even with extensive hyperparameter search. This might be a direct result of insufficient inner optimization steps (we choose 10 steps). However, more inner optimization steps are not acceptable, as currently these algorithms already consume 3 to 10 more running time compared to `OMD`, `EGPO`, and `OMWU`.

• The duality gap of `OMWU` (and occasionally of other algorithms) exhibits noise. This highlights the necessity of analyzing $D_{\mathrm{KL}}(\pi^*\|\hat{\pi}^{(t)})$ rather than relying solely on the duality gap.

## 6 CONCLUSION

We have analyzed the `OMWU` algorithm in the context of NLHF. Compared with Wei et al. (2020), our results relax the uniqueness assumption, improve the convergence rate and burn-in characterization, and eliminate the exponential dependence on $D_{\mathrm{KL}}(\pi^*\|\hat{\pi}_1)/\varepsilon$.

This work provides a theoretical guarantee for non-regularized preference-based learning, highlighting the potential of `OMWU` as an alternative to regularizer-dependent methods. However, we are currently not able to reproduce our result on a fine-tuning problem of large language models due to resource constraints.

However, one limitation of our work is that the result still does not hold for any preference matrix. In addition, the burn-in time and convergence rate are not proved to be of tightest order given the related terms (and we believe they are not tight). We leave the generalization and order problem to future research.

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

# Appendix

## A  PROOFS OF LEMMAS ON KL PROJECTION

This appendix establishes several basic properties of the KL projection $p(\boldsymbol{\pi})$.

**Lemma 3.** *Under Assumption 1, we have $\boldsymbol{\pi} \in \mathbb{M}$ if and only if $\boldsymbol{P}\boldsymbol{\pi} = \boldsymbol{0}$.*

*Proof.*  We only prove the "only if" direction, as the reverse is immediate.

By Assumption 1, there exists some $\boldsymbol{\pi}' \in \mathbb{M}$ such that $\pi_a' > 0$ for all $a \in \mathbb{A}$. Since both $\boldsymbol{\pi}$ and $\boldsymbol{\pi}'$ are equilibria, we have

$$\boldsymbol{\pi}'^{\top} \boldsymbol{P}\boldsymbol{\pi} = 0.$$

Expanding, we obtain

$$\sum_{a \in \mathbb{A}} \pi_a' (\boldsymbol{P}\boldsymbol{\pi})_a = 0.$$

Because $\boldsymbol{\pi} \in \mathbb{M}$, each $(\boldsymbol{P}\boldsymbol{\pi})_a \leqslant 0$. Moreover, $\pi_a' > 0$ for all $a \in \mathbb{A}$ by our choice of $\boldsymbol{\pi}'$. These two facts force $(\boldsymbol{P}\boldsymbol{\pi})_a = 0$ for all $a \in \mathbb{A}$, i.e., $\boldsymbol{P}\boldsymbol{\pi} = \boldsymbol{0}$.  □

**Lemma 4.** *Under Assumption 1, for any policy $\boldsymbol{\pi}$ with full support, there exists a unique equilibrium $\boldsymbol{\pi}' \in \mathbb{M}$ such that*

$$\boldsymbol{\pi}' = \arg\min_{\boldsymbol{\pi}'' \in \mathbb{M}} D_{\mathrm{KL}}(\boldsymbol{\pi}'' \| \boldsymbol{\pi}).$$

*Moreover, the minimizer satisfies $\pi_a' > 0$ for all $a \in \mathbb{A}$.*

*Proof.*  By Lemma 3, the set $\mathbb{M}$ is compact. Since $\boldsymbol{\pi}' \mapsto D_{\mathrm{KL}}(\boldsymbol{\pi}' \| \boldsymbol{\pi})$ is continuous on $\mathbb{M}$, existence follows.

To show positivity of all coordinates, note that there exists some $\boldsymbol{\pi}'' \in \mathbb{M}$ with $\pi_a'' > 0$ for every $a \in \mathbb{A}$. Consider

$$f(\lambda) = D_{\mathrm{KL}}\big(\lambda\boldsymbol{\pi}'' + (1 - \lambda)\boldsymbol{\pi}' \,\big\|\, \boldsymbol{\pi}\big), \qquad 0 \leqslant \lambda \leqslant 1.$$

Since $\mathbb{M}$ is convex, $\lambda\boldsymbol{\pi}'' + (1 - \lambda)\boldsymbol{\pi}' \in \mathbb{M}$. Minimality of $\boldsymbol{\pi}'$ implies that $f(\lambda)$ attains its minimum at $\lambda = 0$. Differentiating,

$$f'(\lambda) = \sum_{a \in \mathbb{A}} (\pi_a'' - \pi_a') \log \frac{\lambda\pi_a'' + (1 - \lambda)\pi_a'}{\pi_a}.$$

If $\pi_a' > 0$, the limit as $\lambda \to 0^+$ is finite. If $\pi_a' = 0$, then $\pi_a'' - \pi_a' > 0$, and the limit becomes $-\infty$, contradicting minimality. Hence $\pi_a' > 0$ for all $a$.

For uniqueness, suppose $\boldsymbol{\pi}'$ and $\boldsymbol{\pi}''$ are two minimizers with $D_{\mathrm{KL}}(\boldsymbol{\pi}' \| \boldsymbol{\pi}) = D_{\mathrm{KL}}(\boldsymbol{\pi}'' \| \boldsymbol{\pi})$. Define

$$f(\lambda) = D_{\mathrm{KL}}\big(\lambda\boldsymbol{\pi}' + (1 - \lambda)\boldsymbol{\pi}'' \,\big\|\, \boldsymbol{\pi}\big).$$

Differentiating,

$$f'(\lambda) = \sum_{a \in \mathbb{A}} (\pi_a' - \pi_a'') \log \frac{\lambda\pi_a' + (1 - \lambda)\pi_a''}{\pi_a},$$

$$f''(\lambda) = \sum_{a \in \mathbb{A}} \frac{(\pi_a' - \pi_a'')^2}{\lambda\pi_a' + (1 - \lambda)\pi_a''} \geqslant 0.$$

Thus $f$ is convex. Since $f(0) = f(1)$ achieves the minimum, $f$ must be constant, forcing $f''(\lambda) \equiv 0$. Therefore $\boldsymbol{\pi}' = \boldsymbol{\pi}''$.  □

**Lemma 5.** *Under Assumption 1, we have*

$$p(\hat{\boldsymbol{\pi}}^{(1)}) = p(\hat{\boldsymbol{\pi}}^{(2)}) = \cdots = p(\hat{\boldsymbol{\pi}}^{(t)}) = p(\hat{\boldsymbol{\pi}}^{(t+1)}) = \cdots .$$

*Proof.* From the update rule in Equation (2), there exists a constant $\hat{M}^{(t+1)}$ such that

$$\log \hat{\boldsymbol{\pi}}^{(t+1)} = \log \hat{\boldsymbol{\pi}}^{(t)} + \eta \boldsymbol{P}\boldsymbol{\pi}^{(t)} + \hat{M}^{(t+1)}, \qquad t \geqslant 1.$$

Let $\boldsymbol{\pi}^* \in \mathbb{M}$ be any Nash equilibrium. Then

$$\begin{aligned} D_{\mathrm{KL}}(\boldsymbol{\pi}^* \| \hat{\boldsymbol{\pi}}^{(t+1)}) &= \langle \boldsymbol{\pi}^*, \log \boldsymbol{\pi}^* \rangle - \langle \boldsymbol{\pi}^*, \log \hat{\boldsymbol{\pi}}^{(t+1)} \rangle \\ &= D_{\mathrm{KL}}(\boldsymbol{\pi}^* \| \hat{\boldsymbol{\pi}}^{(t)}) - \eta \langle \boldsymbol{\pi}^*, \boldsymbol{P}\boldsymbol{\pi}^{(t)} \rangle - \langle \boldsymbol{\pi}^*, \hat{M}^{(t+1)} \rangle. \end{aligned}$$

Since $\langle \boldsymbol{\pi}^*, \boldsymbol{P}\boldsymbol{\pi}^{(t)} \rangle = -\langle \boldsymbol{P}\boldsymbol{\pi}^*, \boldsymbol{\pi}^{(t)} \rangle = 0$, the difference reduces to $-\hat{M}^{(t+1)}$, a normalization term. Thus the projection target does not change with $t$, i.e.,

$$p(\hat{\boldsymbol{\pi}}^{(t+1)}) = p(\hat{\boldsymbol{\pi}}^{(t)}).$$

$\square$

# B PROOF OF MATRIX-DEPENDENT CONSTANT

This section establishes that $C_{\boldsymbol{P}}$ is well defined and strictly positive.

**Lemma 6.** *Let $\mathbb{N} \subseteq \mathbb{R}^n$ be a subspace. Then there exists a constant $C(\mathbb{N}) < 1$ such that the following holds: if $\boldsymbol{a}, \boldsymbol{b} \in \mathbb{R}^n$ and $\boldsymbol{r} \in \mathbb{N}$ satisfy $\mathrm{sgn}(\boldsymbol{a}) = \mathrm{sgn}(\boldsymbol{b})$ and $\boldsymbol{b} \in \mathbb{N}^\perp$, then*

$$|\langle \boldsymbol{r}, \boldsymbol{a} \rangle| \leqslant C(\mathbb{N}) \|\boldsymbol{r}\|_2 \|\boldsymbol{a}\|_2.$$

*Proof.* Since $\mathrm{sgn}(\boldsymbol{a}) \in \{-1, 0, 1\}^n$ has only finitely many possibilities, it suffices to fix any nonzero sign pattern $\boldsymbol{s} \in \{-1, 0, 1\}^n$ and show the claim holds with some $C(\mathbb{N}, \boldsymbol{s}) < 1$ under $\mathrm{sgn}(\boldsymbol{a}) = \mathrm{sgn}(\boldsymbol{b}) = \boldsymbol{s}$. By scaling, we may assume $\|\boldsymbol{r}\|_2 = \|\boldsymbol{a}\|_2 = 1$.

If no $\boldsymbol{b} \in \mathbb{N}^\perp$ has sign pattern $\boldsymbol{s}$, then the claim is vacuous and we may take $C(\mathbb{N}, \boldsymbol{s}) = 0$. Otherwise, fix such a vector $\boldsymbol{b} \in \mathbb{N}^\perp$ with $\mathrm{sgn}(\boldsymbol{b}) = \boldsymbol{s}$.

Define

$$\mathbb{S} = \Big\{ \boldsymbol{a} \in \mathbb{R}^n : \|\boldsymbol{a}\|_2 = 1,\ a_i \geqslant 0 \text{ if } s_i = 1,\ a_i = 0 \text{ if } s_i = 0,\ a_i \leqslant 0 \text{ if } s_i = -1 \Big\}.$$

By construction, $\langle \boldsymbol{a}, \boldsymbol{b} \rangle \geqslant 0$ for all $\boldsymbol{a} \in \mathbb{S}$. Moreover, equality cannot hold because $\boldsymbol{a} \neq 0$ and every nonzero coordinate of $\boldsymbol{a}$ agrees in sign with $\boldsymbol{b}$. Thus $\mathbb{S}$ does not intersect $\mathbb{N}$ (since $\boldsymbol{b} \in \mathbb{N}^\perp$).

It follows that for every $\boldsymbol{r} \in \mathbb{N}$ and $\boldsymbol{a} \in \mathbb{S}$ with $\|\boldsymbol{r}\|_2 = 1$, we must have $|\langle \boldsymbol{r}, \boldsymbol{a} \rangle| < 1$. Compactness of $\mathbb{S}$ then ensures

$$C(\mathbb{N}, \boldsymbol{s}) = \max_{\substack{\boldsymbol{r} \in \mathbb{N},\ \|\boldsymbol{r}\|_2 = 1 \\ \boldsymbol{a} \in \mathbb{S}}} |\langle \boldsymbol{r}, \boldsymbol{a} \rangle| < 1.$$

Taking $C(\mathbb{N}) = \max_{\boldsymbol{s}} C(\mathbb{N}, \boldsymbol{s})$ proves the claim. $\square$

We now show that

$$C_{\boldsymbol{P}} = \min_{\boldsymbol{\pi}} \frac{\|\boldsymbol{P}\boldsymbol{\pi}\|_\infty}{\|\boldsymbol{\pi} - p(\boldsymbol{\pi})\|_1}$$

is well defined and positive under Assumption 1.

—

Let

$$\mathbb{N} = \Big\{ \boldsymbol{r} \in \mathbb{R}^{|\mathbb{A}|} : \boldsymbol{P}\boldsymbol{r} = \boldsymbol{0},\ \sum_{a \in \mathbb{A}} r_a = 0 \Big\},$$

and let $\boldsymbol{\pi}'$ denote the orthogonal projection of $\boldsymbol{\pi}$ onto $p(\boldsymbol{\pi}) + \mathbb{N}$, i.e.,

$$\boldsymbol{\pi}' = \operatorname*{arg\,min}_{\boldsymbol{\pi}' \in p(\boldsymbol{\pi}) + \mathbb{N}} \|\boldsymbol{\pi} - \boldsymbol{\pi}'\|_2,$$

where $\boldsymbol{\pi}'$ need not be a distribution. Since $p(\boldsymbol{\pi})_a > 0$ for all $a \in \mathbb{A}$ by Lemma 1, for any $\boldsymbol{r} \in \mathbb{N}$ the perturbed strategy $p(\boldsymbol{\pi}) + \lambda \boldsymbol{r}$ is also a Nash equilibrium for sufficiently small $|\lambda|$ by Lemma 3. By first-order optimality of $p(\boldsymbol{\pi})$, we obtain

$$\langle \boldsymbol{r}, \log \boldsymbol{\pi} - \log p(\boldsymbol{\pi}) \rangle = 0, \quad \forall \boldsymbol{r} \in \mathbb{N},$$

which implies

$$\log \boldsymbol{\pi} - \log p(\boldsymbol{\pi}) \in \mathbb{N}^{\perp}.$$

Since

$$\mathrm{sgn}(\log \boldsymbol{\pi} - \log p(\boldsymbol{\pi})) = \mathrm{sgn}(\boldsymbol{\pi} - p(\boldsymbol{\pi})),$$

Lemma 6 guarantees the existence of $C(\mathbb{N}) < 1$ such that

$$|\langle \boldsymbol{\pi}' - p(\boldsymbol{\pi}), \, \boldsymbol{\pi} - p(\boldsymbol{\pi}) \rangle| \leqslant C(\mathbb{N}) \, \|\boldsymbol{\pi}' - p(\boldsymbol{\pi})\|_2 \, \|\boldsymbol{\pi} - p(\boldsymbol{\pi})\|_2.$$

Since $\boldsymbol{\pi}'$ is the projection of $\boldsymbol{\pi}$ onto $p(\boldsymbol{\pi}) + \mathbb{N}$, we also have

$$\langle \boldsymbol{\pi}' - p(\boldsymbol{\pi}), \, \boldsymbol{\pi} - \boldsymbol{\pi}' \rangle = 0.$$

Combining these facts yields

$$\|\boldsymbol{\pi}' - p(\boldsymbol{\pi})\|_2 \leqslant C(\mathbb{N}) \, \|\boldsymbol{\pi} - p(\boldsymbol{\pi})\|_2,$$

and by the Pythagorean theorem,

$$\|\boldsymbol{\pi} - \boldsymbol{\pi}'\|_2 = \sqrt{\|\boldsymbol{\pi} - p(\boldsymbol{\pi})\|_2^2 - \|\boldsymbol{\pi}' - p(\boldsymbol{\pi})\|_2^2} \geqslant \sqrt{1 - C(\mathbb{N})^2} \, \|\boldsymbol{\pi} - p(\boldsymbol{\pi})\|_2.$$

Next, we relate $\|\boldsymbol{\pi} - \boldsymbol{\pi}'\|_2$ to $\|\boldsymbol{P}\boldsymbol{\pi}\|_{\infty}$. Note that $\boldsymbol{\pi} - \boldsymbol{\pi}' \in \mathbb{N}' \cap \mathbb{N}^{\perp}$, where $\mathbb{N}' = \{\boldsymbol{r} : \sum_{a \in \mathbb{A}} r_a = 0\}$. Since $\mathbb{N}$ is the null space of $\boldsymbol{P}|_{\mathbb{N}'}$, elementary linear algebra gives

$$\|\boldsymbol{P}(\boldsymbol{\pi} - \boldsymbol{\pi}')\|_2 \geqslant \lambda_{\min} \|\boldsymbol{\pi} - \boldsymbol{\pi}'\|_2,$$

where $\lambda_{\min}$ is the smallest positive singular value of $\boldsymbol{P}|_{\mathbb{N}'}$.

Finally, combining all inequalities:

$$\begin{aligned}
\|\boldsymbol{P}\boldsymbol{\pi}\|_{\infty} &\geqslant \sqrt{|\mathbb{A}|} \, \|\boldsymbol{P}\boldsymbol{\pi}\|_2 \\
&\geqslant \lambda_{\min} \sqrt{|\mathbb{A}|} \, \|\boldsymbol{\pi} - \boldsymbol{\pi}'\|_2 \\
&\geqslant \lambda_{\min} \sqrt{|\mathbb{A}|(1 - C(\mathbb{N})^2)} \, \|\boldsymbol{\pi} - p(\boldsymbol{\pi})\|_2 \\
&\geqslant \lambda_{\min} |\mathbb{A}| \sqrt{1 - C(\mathbb{N})^2} \, \|\boldsymbol{\pi} - p(\boldsymbol{\pi})\|_1.
\end{aligned}$$

Thus we may take

$$C_{\boldsymbol{P}} = \lambda_{\min} |\mathbb{A}| \sqrt{1 - C(\mathbb{N})^2} > 0.$$

## C    PROOF OF THE MAIN THEOREM

### C.1    BASIC LEMMAS

**Lemma 7** (Pinsker's Inequality)**.** *If $\boldsymbol{\pi}$ and $\boldsymbol{\pi}'$ are probability distributions, then*

$$D_{\mathrm{KL}}(\boldsymbol{\pi} \| \boldsymbol{\pi}') \geqslant \tfrac{1}{2} \|\boldsymbol{\pi} - \boldsymbol{\pi}'\|_1^2.$$

**Lemma 8.** *Let $\{x_n\}_{n=0}^{t}$ be a decreasing sequence. Suppose there exists a nonnegative, non-increasing, continuously differentiable function $f$ on $[x_t, x_0]$ such that*

$$x_n - x_{n+1} \geqslant f(x_n)$$

*for all $n = 0, 1, \ldots, t - 1$. Then*

$$\int_{x_t}^{x_0} \frac{1}{f(x)} \, \mathrm{d}x \geqslant t.$$

*Proof.* Since $f$ is non-increasing, we have

$$1 \leqslant \frac{x_n - x_{n+1}}{f(x_n)} = \int_{x_{n+1}}^{x_n} \frac{\mathrm{d}x}{f(x_n)} \leqslant \int_{x_{n+1}}^{x_n} \frac{\mathrm{d}x}{f(x)}$$

for all $n = 0, 1, \ldots, t-1$. Summing over $n$ gives

$$t \leqslant \int_{x_t}^{x_0} \frac{\mathrm{d}x}{f(x)}.$$

$\square$

**Lemma 9.** *Let $\{x_n\}_{n=0}^t$ be a decreasing sequence. Suppose there exists a nonnegative, non-decreasing, continuously differentiable function $f$ on $[x_t, x_0]$ such that*

$$x_n - x_{n+1} \geqslant f(x_{n+1})$$

*for all $n = 0, 1, \ldots, t-1$. Then*

$$\ln \frac{f(x_0)}{f(x_t)} + \int_{x_t}^{x_0} \frac{1}{f(x)} \, \mathrm{d}x \geqslant t.$$

*Proof.* Let $F(x) = x + f(x)$. Then $F'(x) = 1 + f'(x) > 0$, so $F$ is strictly increasing and has an inverse $F^{-1}$ on $[F(x_t), F(x_0)]$. Denote $y_n = F(x_n)$. Then

$$y_{n+1} \leqslant y_n - f(x_n) = y_n - f(F^{-1}(y_n))$$

for $n = 0, 1, \ldots, t-1$. Applying Lemma 8 to the sequence $\{y_n\}$ yields

$$t \leqslant \int_{F(x_t)}^{F(x_0)} \frac{\mathrm{d}y}{f(F^{-1}(y))}.$$

Now substitute $y = F(x) = x + f(x)$. Then $\mathrm{d}y = (1 + f'(x))\mathrm{d}x$, so

$$\int_{F(x_t)}^{F(x_0)} \frac{\mathrm{d}y}{f(F^{-1}(y))} = \int_{x_t}^{x_0} \frac{1 + f'(x)}{f(x)} \, \mathrm{d}x = \int_{x_t}^{x_0} \frac{1}{f(x)} \, \mathrm{d}x + \int_{f(x_t)}^{f(x_0)} \frac{\mathrm{d}t}{t}.$$

The second term equals $\ln \frac{f(x_0)}{f(x_t)}$, giving the claim. $\square$

### C.2 Monotonicity of $\Theta_t$

Define

$$\Theta_t = D_{\mathrm{KL}}(\boldsymbol{\pi}^* \| \hat{\boldsymbol{\pi}}^{(t)}) + 4\eta^2 L^2 \, D_{\mathrm{KL}}(\hat{\boldsymbol{\pi}}^{(t)} \| \boldsymbol{\pi}^{(t-1)}),$$

where $L = \max_{a,b \in \mathbb{A}} P_{a,b}$.

**Lemma 10.**

$$\Theta_t - \Theta_{t+1} \geqslant \left(1 - 4\eta^2 L^2\right)\left(D_{\mathrm{KL}}(\boldsymbol{\pi}^{(t)} \| \hat{\boldsymbol{\pi}}^{(t)}) + D_{\mathrm{KL}}(\hat{\boldsymbol{\pi}}^{(t+1)} \| \boldsymbol{\pi}^{(t)})\right).$$

*In particular, if $\eta L < \frac{1}{2}$, then $\Theta_t$ is strictly decreasing.*

*Proof.* From the update rule Equation (2),

$$\langle \boldsymbol{\pi}^* - \boldsymbol{\pi}^{(t)}, \log \hat{\boldsymbol{\pi}}^{(t+1)} \rangle = \langle \boldsymbol{\pi}^* - \boldsymbol{\pi}^{(t)}, \log \hat{\boldsymbol{\pi}}^{(t)} + \eta \boldsymbol{P} \boldsymbol{\pi}^{(t)} \rangle.$$

Since $\boldsymbol{\pi}^*$ is an equilibrium and $\boldsymbol{P}$ is skew-symmetric,

$$\langle \boldsymbol{\pi}^*, \boldsymbol{P} \boldsymbol{\pi}^{(t)} \rangle = -\langle \boldsymbol{P} \boldsymbol{\pi}^*, \boldsymbol{\pi}^{(t)} \rangle \geqslant 0, \quad \langle \boldsymbol{\pi}^{(t)}, \boldsymbol{P} \boldsymbol{\pi}^{(t)} \rangle = 0,$$

hence

$$\langle \boldsymbol{\pi}^* - \boldsymbol{\pi}^{(t)}, \log \hat{\boldsymbol{\pi}}^{(t+1)} \rangle = \langle \boldsymbol{\pi}^* - \boldsymbol{\pi}^{(t)}, \log \hat{\boldsymbol{\pi}}^{(t)} \rangle.$$

Therefore,

$$D_{\mathrm{KL}}(\boldsymbol{\pi}^* \| \hat{\boldsymbol{\pi}}^{(t)}) - D_{\mathrm{KL}}(\boldsymbol{\pi}^* \| \hat{\boldsymbol{\pi}}^{(t+1)}) = \langle \boldsymbol{\pi}^*, \log \hat{\boldsymbol{\pi}}^{(t+1)} \rangle - \langle \boldsymbol{\pi}^*, \log \hat{\boldsymbol{\pi}}^{(t)} \rangle$$

$$\geqslant \langle \boldsymbol{\pi}^{(t)}, \log \hat{\boldsymbol{\pi}}^{(t+1)} - \log \hat{\boldsymbol{\pi}}^{(t)} \rangle.$$

Subtracting $D_{\mathrm{KL}}(\hat{\boldsymbol{\pi}}^{(t+1)} \| \boldsymbol{\pi}^{(t)})$ and rearranging gives

$$D_{\mathrm{KL}}(\boldsymbol{\pi}^* \| \hat{\boldsymbol{\pi}}^{(t)}) - D_{\mathrm{KL}}(\boldsymbol{\pi}^* \| \hat{\boldsymbol{\pi}}^{(t+1)}) - D_{\mathrm{KL}}(\hat{\boldsymbol{\pi}}^{(t+1)} \| \boldsymbol{\pi}^{(t)})$$
$$\geqslant \langle \boldsymbol{\pi}^{(t)} - \hat{\boldsymbol{\pi}}^{(t+1)}, \log \hat{\boldsymbol{\pi}}^{(t+1)} - \log \boldsymbol{\pi}^{(t)} \rangle + D_{\mathrm{KL}}(\boldsymbol{\pi}^{(t)} \| \hat{\boldsymbol{\pi}}^{(t)}).$$

Meanwhile, by Lemma 7,

$$\|\boldsymbol{\pi}^{(t)} - \hat{\boldsymbol{\pi}}^{(t+1)}\|_1^2 \leqslant D_{\mathrm{KL}}(\boldsymbol{\pi}^{(t)} \| \hat{\boldsymbol{\pi}}^{(t+1)}) + D_{\mathrm{KL}}(\hat{\boldsymbol{\pi}}^{(t+1)} \| \boldsymbol{\pi}^{(t)})$$
$$= \langle \boldsymbol{\pi}^{(t)} - \hat{\boldsymbol{\pi}}^{(t+1)}, \log \boldsymbol{\pi}^{(t)} - \log \hat{\boldsymbol{\pi}}^{(t+1)} \rangle$$
$$= \langle \boldsymbol{\pi}^{(t)} - \hat{\boldsymbol{\pi}}^{(t+1)}, \eta \boldsymbol{P}(\boldsymbol{\pi}^{(t-1)} - \boldsymbol{\pi}^{(t)}) \rangle$$
$$\leqslant \eta \|\boldsymbol{\pi}^{(t)} - \hat{\boldsymbol{\pi}}^{(t+1)}\|_1 \|\boldsymbol{P}(\boldsymbol{\pi}^{(t)} - \boldsymbol{\pi}^{(t-1)})\|_\infty$$
$$\leqslant \eta L \|\boldsymbol{\pi}^{(t)} - \hat{\boldsymbol{\pi}}^{(t+1)}\|_1 \|\boldsymbol{\pi}^{(t)} - \boldsymbol{\pi}^{(t-1)}\|_1.$$

Thus,

$$\|\boldsymbol{\pi}^{(t)} - \hat{\boldsymbol{\pi}}^{(t+1)}\|_1 \leqslant \eta L \|\boldsymbol{\pi}^{(t)} - \boldsymbol{\pi}^{(t-1)}\|_1.$$

Plugging back,

$$|\langle \boldsymbol{\pi}^{(t)} - \hat{\boldsymbol{\pi}}^{(t+1)}, \log \hat{\boldsymbol{\pi}}^{(t+1)} - \log \boldsymbol{\pi}^{(t)} \rangle| \leqslant \eta^2 L^2 \|\boldsymbol{\pi}^{(t)} - \boldsymbol{\pi}^{(t-1)}\|_1^2$$
$$\leqslant 2\eta^2 L^2 \big( \|\boldsymbol{\pi}^{(t)} - \hat{\boldsymbol{\pi}}^{(t)}\|_1^2 + \|\hat{\boldsymbol{\pi}}^{(t)} - \boldsymbol{\pi}^{(t-1)}\|_1^2 \big)$$
$$\leqslant 4\eta^2 L^2 \big( D_{\mathrm{KL}}(\boldsymbol{\pi}^{(t)} \| \hat{\boldsymbol{\pi}}^{(t)}) + D_{\mathrm{KL}}(\hat{\boldsymbol{\pi}}^{(t)} \| \boldsymbol{\pi}^{(t-1)}) \big).$$

Combining with the earlier inequality proves the claim. $\qquad\square$

### C.3 OTHER IMPORTANT INEQUALITIES

Let
$$M^{(t)} = \log \langle \hat{\boldsymbol{\pi}}^{(t)}, \exp(\eta \boldsymbol{P} \boldsymbol{\pi}^{(t-1)}) \rangle, \quad \hat{M}^{(t+1)} = \log \langle \hat{\boldsymbol{\pi}}^{(t)}, \exp(\eta \boldsymbol{P} \boldsymbol{\pi}^{(t)}) \rangle$$
be the normalizing constants chosen so that

$$\log \boldsymbol{\pi}^{(t)} = \log \hat{\boldsymbol{\pi}}^{(t)} + \eta \boldsymbol{P} \boldsymbol{\pi}^{(t-1)} - M^{(t)}, \quad \log \hat{\boldsymbol{\pi}}^{(t+1)} = \log \hat{\boldsymbol{\pi}}^{(t)} + \eta \boldsymbol{P} \boldsymbol{\pi}^{(t)} - \hat{M}^{(t+1)}.$$

Also, denote
$$K^{(t)} = \max_{a \in \mathbb{A}} \hat{\boldsymbol{\pi}}_a^{(t)} |\eta \boldsymbol{P} \boldsymbol{\pi}^{(t-1)}|_a, \quad \hat{K}^{(t+1)} = \max_{a \in \mathbb{A}} \hat{\boldsymbol{\pi}}_a^{(t)} |\eta \boldsymbol{P} \boldsymbol{\pi}^{(t)}|_a.$$

**Lemma 11.**
$$\| \log \hat{\boldsymbol{\pi}}^{(t+1)} - \log \hat{\boldsymbol{\pi}}^{(t)} \|_\infty \leqslant 2\eta L.$$

*Proof.* Recall the update rule Equation (2):

$$\log \hat{\boldsymbol{\pi}}^{(t+1)} = \log \hat{\boldsymbol{\pi}}^{(t)} + \eta \boldsymbol{P} \boldsymbol{\pi}^{(t)} - \hat{M}^{(t+1)}. \tag{3}$$

From the definition of $\hat{M}^{(t+1)}$, we have $|\hat{M}^{(t+1)}| \leqslant \|\boldsymbol{P} \boldsymbol{\pi}^{(t)}\|_\infty$. Hence

$$\| \log \hat{\boldsymbol{\pi}}^{(t+1)} - \log \hat{\boldsymbol{\pi}}^{(t)} \|_\infty \leqslant 2\|\eta \boldsymbol{P} \boldsymbol{\pi}^{(t)}\|_\infty \leqslant 2\eta L.$$

$\qquad\square$

**Lemma 12.**
$$|\hat{M}^{(t+1)}| \leqslant \mathrm{e}^{\eta L} |\mathbb{A}| \hat{K}^{(t+1)}, \quad |M^{(t)}| \leqslant \mathrm{e}^{\eta L} |\mathbb{A}| K^{(t)}.$$

*Proof.* From the convexity of the exponential function, we have

$$1 + x \leqslant e^x \leqslant 1 + \frac{e^{\eta L} - 1}{\eta L} x \quad \text{for all } x \in [0, \eta L].$$

Since $\eta(\boldsymbol{P}\boldsymbol{\pi}^{(t)})_a \in [-\eta L, \eta L]$, it follows that

$$1 + \eta(\boldsymbol{P}\boldsymbol{\pi}^{(t)})_a \leqslant \exp(\eta(\boldsymbol{P}\boldsymbol{\pi}^{(t)})_a) \leqslant 1 + \frac{e^{\eta L} - 1}{\eta L} \max\{\eta(\boldsymbol{P}\boldsymbol{\pi}^{(t)})_a, 0\}.$$

Recall that

$$\exp(\hat{M}^{(t+1)}) = \sum_{a \in \mathbb{A}} \hat{\pi}_a^{(t)} \exp(\eta(\boldsymbol{P}\boldsymbol{\pi}^{(t)})_a).$$

Hence

$$1 + |\mathbb{A}|\eta \min_{a \in \mathbb{A}} \hat{\pi}_a^{(t)}(\boldsymbol{P}\boldsymbol{\pi}^{(t)})_a \leqslant e^{\hat{M}^{(t+1)}} \leqslant 1 + \frac{e^{\eta L} - 1}{\eta L}|\mathbb{A}| \max\{\eta \max_{a \in \mathbb{A}} \hat{\pi}_a^{(t)}(\boldsymbol{P}\boldsymbol{\pi}^{(t)})_a, 0\}.$$

Thus,

$$\hat{K}^{(t+1)} \geqslant \frac{\hat{M}^{(t+1)}}{e^{\eta L}|\mathbb{A}|}.$$

The other inequality follows analogously. $\square$

**Lemma 13.**

$$\|\hat{\boldsymbol{\pi}}^{(t+1)} - \hat{\boldsymbol{\pi}}^{(t)}\|_1 \leqslant \frac{(e^{2\eta L} - 1)(e^{\eta L} + 1)|\mathbb{A}|\hat{K}^{(t+1)}}{2\eta L}.$$

*Also,*

$$\|\boldsymbol{\pi}^{(t)} - \hat{\boldsymbol{\pi}}^{(t)}\|_1 \leqslant \frac{(e^{2\eta L} - 1)(e^{\eta L} + 1)|\mathbb{A}|K^{(t)}}{2\eta L}.$$

*Proof.* From the update rule Equation (2),

$$\hat{\pi}_a^{(t+1)} - \hat{\pi}_a^{(t)} = \hat{\pi}_a^{(t)} \exp((\eta\boldsymbol{P}\boldsymbol{\pi}^{(t)})_a - \hat{M}^{(t+1)}) - \hat{\pi}_a^{(t)}.$$

When $(\eta\boldsymbol{P}\boldsymbol{\pi}^{(t)})_a - \hat{M}^{(t+1)} \geqslant 0$, convexity and the trivial bound $|(\eta\boldsymbol{P}\boldsymbol{\pi}^{(t)})_a - \hat{M}^{(t+1)}| \leqslant 2\eta L$ give

$$\exp((\eta\boldsymbol{P}\boldsymbol{\pi}^{(t)})_a - \hat{M}^{(t+1)}) \leqslant 1 + \frac{e^{2\eta L} - 1}{2\eta L}((\eta\boldsymbol{P}\boldsymbol{\pi}^{(t)})_a - \hat{M}^{(t+1)}).$$

Hence,

$$0 \leqslant \hat{\pi}_a^{(t+1)} - \hat{\pi}_a^{(t)} \leqslant \frac{(e^{2\eta L} - 1)(\hat{\pi}_a^{(t)}|\hat{M}^{(t+1)}| + \hat{K}^{(t+1)})}{2\eta L}.$$

When $(\eta\boldsymbol{P}\boldsymbol{\pi}^{(t)})_a - \hat{M}^{(t+1)} \leqslant 0$, we use

$$\exp((\eta\boldsymbol{P}\boldsymbol{\pi}^{(t)})_a - \hat{M}^{(t+1)}) \geqslant 1 + (\eta\boldsymbol{P}\boldsymbol{\pi}^{(t)})_a - \hat{M}^{(t+1)},$$

which implies

$$0 \geqslant \hat{\pi}_a^{(t+1)} - \hat{\pi}_a^{(t)} \geqslant -\hat{\pi}_a^{(t)}|\hat{M}^{(t+1)}| - \hat{K}^{(t+1)}.$$

Combining the two cases and noting that $e^{2\eta L} \geqslant 1 + 2\eta L$, we obtain

$$\|\hat{\boldsymbol{\pi}}^{(t+1)} - \hat{\boldsymbol{\pi}}^{(t)}\|_1 \leqslant \frac{e^{2\eta L} - 1}{2\eta L}\left(|\hat{M}^{(t+1)}| + |\mathbb{A}|\hat{K}^{(t+1)}\right).$$

Applying Lemma 12 yields the claimed bound. The second inequality follows by the same reasoning. $\square$

Recall that $\varepsilon = \min_{a \in \mathbb{A}} \pi_a^*$.

**Lemma 14.** *If $\pi_a > 0$ for all $a \in \mathbb{A}$, then*

$$\varepsilon\|\log \boldsymbol{\pi}^* - \log \boldsymbol{\pi}\|_1 \leqslant D_{\mathrm{KL}}(\boldsymbol{\pi}^*\|\boldsymbol{\pi}) + 2e^{-1}.$$

*Proof.* We first note

$$\varepsilon \| \log \boldsymbol{\pi}^* - \log \boldsymbol{\pi} \|_1 \leqslant \| \boldsymbol{\pi}^* (\log \boldsymbol{\pi}^* - \log \boldsymbol{\pi}) \|_1.$$

Thus it suffices to prove

$$\| \boldsymbol{\pi}^* (\log \boldsymbol{\pi}^* - \log \boldsymbol{\pi}) \|_1 \leqslant D_{\mathrm{KL}}(\boldsymbol{\pi}^* \| \boldsymbol{\pi}) + 2\mathrm{e}^{-1}.$$

Rearranging, this is equivalent to showing

$$\sum_{a \in \mathbb{A} : \pi_a^* < \pi_a} \pi_a^* (\log \pi_a - \log \pi_a^*) \leqslant \mathrm{e}^{-1}.$$

Define

$$S = \sum_{a \in \mathbb{A} : \pi_a^* < \pi_a} \pi_a, \quad S^* = \sum_{a \in \mathbb{A} : \pi_a^* < \pi_a} \pi_a^*.$$

By Jensen's inequality,

$$\sum_{a : \pi_a^* < \pi_a} \pi_a^* (\log \pi_a - \log \pi_a^*) \leqslant S^* \log(S/S^*).$$

Since $S \leqslant 1$, we further have

$$S^* \log(S/S^*) \leqslant -S^* \log S^* \leqslant \mathrm{e}^{-1}.$$

This completes the proof. $\qquad\square$

**Lemma 15.**

$$\| \boldsymbol{\pi}^* - \boldsymbol{\pi} \|_1 \geqslant 2\varepsilon \sqrt{1 - \exp(-D_{\mathrm{KL}}(\boldsymbol{\pi}^* \| \boldsymbol{\pi})/\varepsilon)}.$$

*Proof.* Let

$$Q = \sum_{a \in \mathbb{A}} (\pi_a^* - \pi_a)^- = \sum_{a \in \mathbb{A}} (\pi_a^* - \pi_a)^+ = \frac{1}{2} \| \boldsymbol{\pi}^* - \boldsymbol{\pi} \|_1.$$

Note that for fixed $Q < \varepsilon$, $\pi_a > 0$ for all $a \in \mathbb{A}$. Hence in this regime $D_{\mathrm{KL}}(\boldsymbol{\pi}^* \| \boldsymbol{\pi})$ is finite, so it attains a maximum at some $\boldsymbol{\pi}$. We restrict to such maximizers.

Pick $a \in \mathbb{A}$ with $\pi_a^* = \varepsilon$. If multiple exist, choose $a$ minimizing $\pi_a$. We claim no $a' \neq a$ can satisfy $\pi_{a'} < \pi_{a'}^*$. Suppose such $a'$ exists. Set

$$p = \pi_a^*, \quad q = \pi_{a'}^*, \quad x = \pi_a^* - \pi_a, \quad y = \pi_{a'}^* - \pi_{a'}.$$

Consider replacing $(\pi_a, \pi_{a'})$ with $(\pi_a - (q - \pi_{a'}), q)$. The KL contribution changes from

$$-p \log(p - x) - q \log(q - y)$$

to

$$-p \log(p - x - y) - q \log q.$$

Thus we need to check

$$q \log \frac{q}{q - y} < p \log \frac{p - x}{p - x - y}. \tag{4}$$

Define $f(q) = q \log \frac{q}{q - y}$. Then

$$f'(q) = \log \left(1 + \frac{y}{q - y}\right) - \frac{y}{q - y} < 0,$$

since $q \geqslant p > x + y > y$. Hence $f(q) \leqslant f(p)$. Moreover,

$$f(p) \leqslant p \log \frac{p - x}{p - x - y}$$

is equivalent to $p(p - x - y) \leqslant (p - x)(p - y)$, which holds.

Now, equality in Equation (4) would require simultaneously $f(q) = f(p)$ and $p(p - x - y) = (p - x)(p - y)$. The first condition holds only when $p = q$, and the second only when $x = 0$ (since by assumption $y > 0$). Thus equality would force $p = q$, $x = 0$, and $y > 0$, which contradicts

the choice of $a$: in that case $\pi_a^* = \pi_{a'}^*$, but $\pi_{a'} < \pi_a$. Therefore strict inequality holds, and the KL strictly increases, contradicting maximality.

This proves uniqueness of such $a$. A symmetric argument shows there is also a unique $b \in \mathbb{A}$ with $\pi_b > \pi_b^*$.

Hence

$$D_{\mathrm{KL}}(\boldsymbol{\pi}^* \| \boldsymbol{\pi}) = \pi_a^* \log \frac{\pi_a^*}{\pi_a^* - Q} + \pi_b^* \log \frac{\pi_b^*}{\pi_b^* + Q} \leqslant \varepsilon \log \frac{\varepsilon^2}{\varepsilon^2 - Q^2}.$$

Solving for $Q$ gives

$$Q \geqslant \varepsilon \sqrt{1 - \exp(-D_{\mathrm{KL}}(\boldsymbol{\pi}^* \| \boldsymbol{\pi})/\varepsilon)},$$

which implies the desired bound since $\|\boldsymbol{\pi}^* - \boldsymbol{\pi}\|_1 = 2Q$. $\qquad\square$

### C.4 UNIFIED ANALYSIS OF ALL CASES

**Lemma 16.**

$$\Theta_t - \Theta_{t+1} \geqslant \frac{1 - 4\eta^2 L^2}{(\sqrt{2}\eta L + 2e^{2\eta L})^2} \max\{\hat{K}^{(t+1)}, K^{(t+1)}\}^2$$

*Proof.* From Equation (2), we have

$$\log \hat{\pi}_a^{(t+1)} - \langle \hat{\boldsymbol{\pi}}^{(t+1)}, \log \hat{\boldsymbol{\pi}}^{(t+1)} \rangle = \log \hat{\pi}_a^{(t)} + \eta(\boldsymbol{P}\boldsymbol{\pi}^{(t)})_a - \langle \hat{\boldsymbol{\pi}}^{(t+1)}, \log \hat{\boldsymbol{\pi}}^{(t)} + \eta \boldsymbol{P}\boldsymbol{\pi}^{(t)} \rangle.$$

Rearranging gives

$$\eta(\boldsymbol{P}\boldsymbol{\pi}^{(t)})_a = \langle \eta \boldsymbol{P}\boldsymbol{\pi}^{(t)}, \hat{\boldsymbol{\pi}}^{(t+1)} \rangle + (\log \hat{\pi}_a^{(t+1)} - \log \hat{\pi}_a^{(t)}) - D_{\mathrm{KL}}(\hat{\boldsymbol{\pi}}^{(t+1)} \| \hat{\boldsymbol{\pi}}^{(t)}).$$

Since $\langle \boldsymbol{P}\boldsymbol{\pi}^{(t)}, \boldsymbol{\pi}^{(t)} \rangle = 0$, the first term satisfies

$$\langle \eta \boldsymbol{P}\boldsymbol{\pi}^{(t)}, \hat{\boldsymbol{\pi}}^{(t+1)} \rangle = \langle \eta \boldsymbol{P}\boldsymbol{\pi}^{(t)}, \hat{\boldsymbol{\pi}}^{(t+1)} - \boldsymbol{\pi}^{(t)} \rangle \leqslant \eta L \|\hat{\boldsymbol{\pi}}^{(t+1)} - \boldsymbol{\pi}^{(t)}\|_1.$$

For the second term, using

$$|\log a - \log b| \leqslant \frac{|a - b|}{\min\{a, b\}} \leqslant \frac{\exp(|\log a - \log b|)}{\max\{a, b\}}|a - b|,$$

together with Lemma 11, we obtain

$$|\log \hat{\pi}_a^{(t+1)} - \log \hat{\pi}_a^{(t)}| \leqslant \frac{e^{2\eta L}}{\max\{\hat{\pi}_a^{(t)}, \hat{\pi}_a^{(t+1)}\}} \|\hat{\boldsymbol{\pi}}^{(t+1)} - \hat{\boldsymbol{\pi}}^{(t)}\|_1.$$

Thus,

$$|\eta(\boldsymbol{P}\boldsymbol{\pi}^{(t)})_a| \leqslant \frac{e^{2\eta L}}{\max\{\hat{\pi}_a^{(t)}, \hat{\pi}_a^{(t+1)}\}} \|\hat{\boldsymbol{\pi}}^{(t+1)} - \hat{\boldsymbol{\pi}}^{(t)}\|_1 + \eta L \|\hat{\boldsymbol{\pi}}^{(t+1)} - \boldsymbol{\pi}^{(t)}\|_1.$$

Multiplying both sides by $\max\{\hat{\pi}_a^{(t)}, \hat{\pi}_a^{(t+1)}\}$ and taking the maximum over $a \in \mathbb{A}$, we obtain

$$\max\{\hat{K}^{(t+1)}, K^{(t+1)}\} \leqslant e^{2\eta L} \|\hat{\boldsymbol{\pi}}^{(t+1)} - \hat{\boldsymbol{\pi}}^{(t)}\|_1 + \eta L \|\hat{\boldsymbol{\pi}}^{(t+1)} - \boldsymbol{\pi}^{(t)}\|_1.$$

By Lemma 7,

$$D_{\mathrm{KL}}(\boldsymbol{\pi}^{(t)} \| \hat{\boldsymbol{\pi}}^{(t)}) + D_{\mathrm{KL}}(\hat{\boldsymbol{\pi}}^{(t+1)} \| \boldsymbol{\pi}^{(t)}) \geqslant \tfrac{1}{2}\|\boldsymbol{\pi}^{(t)} - \hat{\boldsymbol{\pi}}^{(t)}\|_1^2 + \tfrac{1}{2}\|\hat{\boldsymbol{\pi}}^{(t+1)} - \boldsymbol{\pi}^{(t)}\|_1^2$$
$$\geqslant \tfrac{1}{4}\|\hat{\boldsymbol{\pi}}^{(t+1)} - \hat{\boldsymbol{\pi}}^{(t)}\|_1^2.$$

Hence,

$$(\sqrt{2}\eta L + 2e^{2\eta L})\sqrt{D_{\mathrm{KL}}(\boldsymbol{\pi}^{(t)} \| \hat{\boldsymbol{\pi}}^{(t)}) + D_{\mathrm{KL}}(\hat{\boldsymbol{\pi}}^{(t+1)} \| \boldsymbol{\pi}^{(t)})}$$
$$\geqslant \eta L \|\hat{\boldsymbol{\pi}}^{(t+1)} - \boldsymbol{\pi}^{(t)}\|_1 + e^{2\eta L} \|\hat{\boldsymbol{\pi}}^{(t+1)} - \hat{\boldsymbol{\pi}}^{(t)}\|_1$$

$$\geqslant \max\{\hat{K}^{(t+1)}, K^{(t+1)}\}.$$

Thus, by Lemma 10,

$$\Theta_t - \Theta_{t+1} \geqslant \frac{1 - 4\eta^2 L^2}{(\sqrt{2}\eta L + 2\mathrm{e}^{2\eta L})^2} \max\{\hat{K}^{(t+1)}, K^{(t+1)}\}^2.$$

$\square$

**Remark 6.** For notational simplicity, define

$$C_1 = \frac{1 - 4\eta^2 L^2}{(\sqrt{2}\eta L + 2\mathrm{e}^{2\eta L})^2}.$$

Recall that

$$\Phi_t = \frac{\langle \log \hat{\boldsymbol{\pi}}^{(t)} - \log \boldsymbol{\pi}^*, \eta \boldsymbol{P}\hat{\boldsymbol{\pi}}^{(t)}\rangle}{(\Theta_t + 2\mathrm{e}^{-1})^2}.$$

Thus,

$$\Phi_t - \Phi_{t+1} = \frac{\langle \log \hat{\boldsymbol{\pi}}^{(t+1)} - \log \boldsymbol{\pi}^*, \eta \boldsymbol{P}\hat{\boldsymbol{\pi}}^{(t)} - \eta \boldsymbol{P}\hat{\boldsymbol{\pi}}^{(t+1)}\rangle + \langle \log \hat{\boldsymbol{\pi}}^{(t)} - \log \hat{\boldsymbol{\pi}}^{(t+1)}, \eta \boldsymbol{P}\hat{\boldsymbol{\pi}}^{(t)}\rangle}{(\Theta_t + 2\mathrm{e}^{-1})^2}$$

$$+ \langle \log \hat{\boldsymbol{\pi}}^{(t+1)} - \log \boldsymbol{\pi}^*, \eta \boldsymbol{P}\hat{\boldsymbol{\pi}}^{(t+1)}\rangle \left(\frac{1}{(\Theta_t + 2\mathrm{e}^{-1})^2} - \frac{1}{(\Theta_{t+1} + 2\mathrm{e}^{-1})^2}\right)$$

$$\geqslant \frac{\langle \log \hat{\boldsymbol{\pi}}^{(t+1)} - \log \boldsymbol{\pi}^*, \eta \boldsymbol{P}\hat{\boldsymbol{\pi}}^{(t)} - \eta \boldsymbol{P}\hat{\boldsymbol{\pi}}^{(t+1)}\rangle + \langle \log \hat{\boldsymbol{\pi}}^{(t)} - \log \hat{\boldsymbol{\pi}}^{(t+1)}, \eta \boldsymbol{P}\hat{\boldsymbol{\pi}}^{(t)}\rangle}{(\Theta_t + 2\mathrm{e}^{-1})^2}$$

$$- \frac{\eta L \|\log \hat{\boldsymbol{\pi}}^{(t+1)} - \log \boldsymbol{\pi}^*\|_1 (\Theta_{t+1} + \Theta_t + 4\mathrm{e}^{-1})}{(\Theta_t + 2\mathrm{e}^{-1})^2 (\Theta_{t+1} + 2\mathrm{e}^{-1})^2} (\Theta_t - \Theta_{t+1}).$$

We bound each term separately.

**Lemma 17.**

$$\langle \log \hat{\boldsymbol{\pi}}^{(t)} - \log \hat{\boldsymbol{\pi}}^{(t+1)}, \eta \boldsymbol{P}\hat{\boldsymbol{\pi}}^{(t)}\rangle \geqslant \frac{\eta^2}{4} \|\boldsymbol{P}\boldsymbol{\pi}^{(t)}\|_\infty^2 - \frac{|\mathbb{A}|^3 \mathrm{e}^{2\eta L}(\hat{K}^{(t+1)})^2}{4}$$

$$- \frac{(\mathrm{e}^{2\eta L} - 1)(\mathrm{e}^{\eta L} + 1)}{2\varepsilon}|\mathbb{A}|(\Theta_t + \Theta_{t+1} + 4\mathrm{e}^{-1})K^{(t)}.$$

*Proof.* From the update rules,

$$\langle \log \hat{\boldsymbol{\pi}}^{(t)} - \log \hat{\boldsymbol{\pi}}^{(t+1)}, \eta \boldsymbol{P}\hat{\boldsymbol{\pi}}^{(t)}\rangle = \langle \log \hat{\boldsymbol{\pi}}^{(t)} - \log \hat{\boldsymbol{\pi}}^{(t+1)}, \eta \boldsymbol{P}\hat{\boldsymbol{\pi}}^{(t)} - \eta \boldsymbol{P}\boldsymbol{\pi}^{(t)}\rangle$$

$$+ \langle \eta \boldsymbol{P}\boldsymbol{\pi}^{(t)} + \hat{M}^{(t+1)}, \eta \boldsymbol{P}\boldsymbol{\pi}^{(t)}\rangle.$$

Since $\eta\|\boldsymbol{P}\boldsymbol{\pi}^{(t)}\|_\infty \geqslant |\hat{M}^{(t+1)}|$, and for some $a \in \mathbb{A}$, $\eta(\boldsymbol{P}\boldsymbol{\pi}^{(t)})_a = \eta\|\boldsymbol{P}\boldsymbol{\pi}^{(t)}\|_\infty$, we obtain

$$\langle \eta \boldsymbol{P}\boldsymbol{\pi}^{(t)} + \hat{M}^{(t+1)}, \eta \boldsymbol{P}\boldsymbol{\pi}^{(t)}\rangle \geqslant (\eta\|\boldsymbol{P}\boldsymbol{\pi}^{(t)}\|_\infty + \hat{M}^{(t+1)}) \cdot \eta\|\boldsymbol{P}\boldsymbol{\pi}^{(t)}\|_\infty - \frac{|\mathbb{A}| - 1}{4}(\hat{M}^{(t+1)})^2$$

$$= \left(\eta\|\boldsymbol{P}\boldsymbol{\pi}^{(t)}\|_\infty + \tfrac{1}{2}\hat{M}^{(t+1)}\right)^2 - \frac{|\mathbb{A}|}{4}(\hat{M}^{(t+1)})^2$$

$$\geqslant \frac{\eta^2}{4}\|\boldsymbol{P}\boldsymbol{\pi}^{(t)}\|_\infty^2 - \frac{|\mathbb{A}|^3 \mathrm{e}^{2\eta L}(\hat{K}^{(t+1)})^2}{4}.$$

Moreover,

$$\langle \log \hat{\boldsymbol{\pi}}^{(t)} - \log \hat{\boldsymbol{\pi}}^{(t+1)}, \eta \boldsymbol{P}\hat{\boldsymbol{\pi}}^{(t)} - \eta \boldsymbol{P}\boldsymbol{\pi}^{(t)}\rangle \geqslant -\eta L(\|\log \hat{\boldsymbol{\pi}}^{(t)} - \log \boldsymbol{\pi}^*\|_1 + \|\log \hat{\boldsymbol{\pi}}^{(t+1)} - \log \boldsymbol{\pi}^*\|_1)\|\hat{\boldsymbol{\pi}}^{(t)} - \boldsymbol{\pi}^{(t)}\|_1$$

$$\geqslant -\frac{(\mathrm{e}^{2\eta L} - 1)(\mathrm{e}^{\eta L} + 1)}{2\varepsilon}|\mathbb{A}|(\Theta_t + \Theta_{t+1} + 4\mathrm{e}^{-1})K^{(t)},$$

where the last step follows from Lemma 13 and Lemma 14. $\square$

**Lemma 18.**

$$\langle \log \hat{\boldsymbol{\pi}}^{(t+1)} - \log \boldsymbol{\pi}^*, \eta \boldsymbol{P} \hat{\boldsymbol{\pi}}^{(t)} - \eta \boldsymbol{P} \hat{\boldsymbol{\pi}}^{(t+1)} \rangle \geqslant -\frac{(\mathrm{e}^{2\eta L} - 1)(\mathrm{e}^{\eta L} + 1)}{2\varepsilon} |\mathbb{A}| (\Theta_{t+1} + 2\mathrm{e}^{-1}) \hat{K}^{(t+1)}.$$

*Proof.* By Lemma 13 and Lemma 14,

$$\langle \log \hat{\boldsymbol{\pi}}^{(t+1)} - \log \boldsymbol{\pi}^*, \eta \boldsymbol{P} \hat{\boldsymbol{\pi}}^{(t)} - \eta \boldsymbol{P} \hat{\boldsymbol{\pi}}^{(t+1)} \rangle \geqslant -\eta L \| \log \hat{\boldsymbol{\pi}}^{(t+1)} - \log \boldsymbol{\pi}^* \|_1 \| \hat{\boldsymbol{\pi}}^{(t)} - \hat{\boldsymbol{\pi}}^{(t+1)} \|_1$$

$$\geqslant -\frac{(\mathrm{e}^{2\eta L} - 1)(\mathrm{e}^{\eta L} + 1)}{2\varepsilon} |\mathbb{A}| (\Theta_{t+1} + 2\mathrm{e}^{-1}) \hat{K}^{(t+1)}.$$

$\square$

**Lemma 19.**

$$\frac{\Theta_{t+1} + \Theta_t + 4\mathrm{e}^{-1}}{(\Theta_t + 2\mathrm{e}^{-1})^2 (\Theta_{t+1} + 2\mathrm{e}^{-1})^2} \leqslant \frac{1}{4} \mathrm{e}^3.$$

*Proof.* Consider $f(u, v) = \frac{u+v}{u^2 v^2}$ for $u, v > 0$. Then

$$\partial_u f(u, v) = -\frac{u + 2v}{u^3 v^2} < 0, \quad \partial_v f(u, v) = -\frac{2u + v}{u^2 v^3} < 0.$$

Thus,

$$f(\Theta_t + 2\mathrm{e}^{-1}, \Theta_{t+1} + 2\mathrm{e}^{-1}) \leqslant f(2\mathrm{e}^{-1}, 2\mathrm{e}^{-1}) = \tfrac{1}{4} \mathrm{e}^3.$$

$\square$

Combining Lemma 17, Lemma 18, Lemma 19, and $\Theta_t \geqslant \Theta_{t+1}$, we obtain

$$\Phi_t - \Phi_{t+1} \geqslant \frac{\eta^2 \| \boldsymbol{P} \boldsymbol{\pi}^{(t)} \|_\infty^2}{4(\Theta_t + 2\mathrm{e}^{-1})^2} - \frac{|\mathbb{A}|^3 \mathrm{e}^{2\eta L} (\hat{K}^{(t+1)})^2}{4(\Theta_t + 2\mathrm{e}^{-1})^2} - \frac{(\mathrm{e}^{2\eta L} - 1)(\mathrm{e}^{\eta L} + 1)}{\varepsilon (\Theta_t + 2\mathrm{e}^{-1})} |\mathbb{A}| K^{(t)}$$

$$- \frac{(\mathrm{e}^{2\eta L} - 1)(\mathrm{e}^{\eta L} + 1)}{2\varepsilon (\Theta_t + 2\mathrm{e}^{-1})} |\mathbb{A}| \hat{K}^{(t+1)} - \tfrac{1}{4} \eta L \mathrm{e}^3 (\Theta_t - \Theta_{t+1}).$$

Define

$$\bar{\Theta}_t = \Theta_t + \frac{4|\mathbb{A}|(\mathrm{e}^{2\eta L} - 1)(\mathrm{e}^{\eta L} + 1)}{\varepsilon \eta^2} (\Theta_t + 2\mathrm{e}^{-1}).$$

**Lemma 20.** *For all $t \geqslant 0$,*

$$\bar{\Theta}_t - \bar{\Theta}_{t+1} \geqslant \frac{\| \boldsymbol{P} \boldsymbol{\pi}^{(t)} \|_\infty^2}{4(\Theta_t + 2\mathrm{e}^{-1})^2} - \frac{|\mathbb{A}|^3 \mathrm{e}^{2\eta L} (\hat{K}^{(t+1)})^2}{4\eta^2 (\Theta_t + 2\mathrm{e}^{-1})^2} - \frac{(\mathrm{e}^{2\eta L} - 1)(\mathrm{e}^{\eta L} + 1)}{2\varepsilon \eta^2 (\Theta_t + 2\mathrm{e}^{-1})} |\mathbb{A}| \hat{K}^{(t+1)}.$$

*Proof.* From the definition of $\bar{\Theta}_t$ and the bound on $\Phi_t - \Phi_{t+1}$,

$$\bar{\Theta}_t - \bar{\Theta}_{t+1} = (\Theta_t - \Theta_{t+1}) + \frac{4|\mathbb{A}|(\mathrm{e}^{2\eta L} - 1)(\mathrm{e}^{\eta L} + 1)}{\varepsilon \eta^2} (\Theta_t - \Theta_{t+1})$$

$$\geqslant (\Theta_t - \Theta_{t+1}) + \frac{4(\Theta_t + 2\mathrm{e}^{-1})(\Phi_t - \Phi_{t+1})}{\eta^2}.$$

Substituting the bound on $\Phi_t - \Phi_{t+1}$ completes the proof. $\square$

Next, we show that $\bar{\Theta}_t$ decreases sufficiently fast in each iteration.

**Lemma 21.** *For all $t \geqslant 0$,*

$$\bar{\Theta}_t - \bar{\Theta}_{t+1} \geqslant \frac{1}{16(\Theta_t + 2\mathrm{e}^{-1})^2} \max \left\{ \| \boldsymbol{P} \boldsymbol{\pi}^{(t)} \|_\infty^2, (\hat{K}^{(t+1)})^2 \right\}.$$

*Proof.* From Lemma 20, it suffices to show

$$\frac{\|\boldsymbol{P}\boldsymbol{\pi}^{(t)}\|_\infty^2}{4(\Theta_t + 2\mathrm{e}^{-1})^2} - \frac{|\mathbb{A}|^3\mathrm{e}^{2\eta L}(\hat{K}^{(t+1)})^2}{4\eta^2(\Theta_t + 2\mathrm{e}^{-1})^2} - \frac{(\mathrm{e}^{2\eta L} - 1)(\mathrm{e}^{\eta L} + 1)}{2\varepsilon\eta^2(\Theta_t + 2\mathrm{e}^{-1})}|\mathbb{A}|\hat{K}^{(t+1)}$$

is at least

$$\frac{1}{16(\Theta_t + 2\mathrm{e}^{-1})^2} \max\left\{\|\boldsymbol{P}\boldsymbol{\pi}^{(t)}\|_\infty^2, (\hat{K}^{(t+1)})^2\right\}.$$

If $\|\boldsymbol{P}\boldsymbol{\pi}^{(t)}\|_\infty \geqslant 2\hat{K}^{(t+1)}$, the negative terms are dominated by the positive part, yielding

$$\bar{\Theta}_t - \bar{\Theta}_{t+1} \geqslant \frac{1}{8(\Theta_t + 2\mathrm{e}^{-1})^2}\|\boldsymbol{P}\boldsymbol{\pi}^{(t)}\|_\infty^2.$$

Otherwise, $\hat{K}^{(t+1)} \geqslant \frac{1}{2}\|\boldsymbol{P}\boldsymbol{\pi}^{(t)}\|_\infty$. Substituting this relation, the inequality simplifies to

$$\bar{\Theta}_t - \bar{\Theta}_{t+1} \geqslant \frac{1}{16(\Theta_t + 2\mathrm{e}^{-1})^2}(\hat{K}^{(t+1)})^2.$$

This completes the proof. □

Combining Lemma 21 with Lemma 8 yields a global bound on the time to reach the region $\Theta_t < \varepsilon$.

**Theorem 3.** *There exists a constant*

$$T_1 = O\left(\frac{\eta^2 L^2|\mathbb{A}|^2(\bar{\Theta}_1^3 + 1)}{(1 - 4\eta^2 L^2)C_{\boldsymbol{P}}^4\varepsilon^6}\right)$$

*such that* $\Theta_{T_1} < \varepsilon$.

*Proof.* If $\Theta_t \geqslant \varepsilon$ for all $t \leqslant T$, then Lemma 21 and Lemma 8 imply

$$T - 1 \leqslant \int_{\bar{\Theta}_T}^{\bar{\Theta}_1} \frac{3(\mathrm{e}^{2\eta L} - 1)^2(\mathrm{e}^{\eta L} + 1)^2(x + 2\mathrm{e}^{-1})^2|\mathbb{A}|^2}{(1 - \mathrm{e}^{-1})C_1 C_{\boldsymbol{P}}^4\varepsilon^6} \, \mathrm{d}x = O\left(\frac{\eta^2 L^2|\mathbb{A}|^2(\bar{\Theta}_1^3 + 1)}{(1 - 4\eta^2 L^2)C_{\boldsymbol{P}}^4\varepsilon^6}\right).$$

Hence such a $T_1$ exists with the stated asymptotic bound. □

Next we handle the phase after $\Theta_t$ enters the small region $\Theta_t < \varepsilon$.

When $\Theta_t < \varepsilon$ we have the lower bound

$$\Theta_t - \Theta_{t+1} \geqslant C_1(\hat{K}^{(t+1)})^2$$

and the trivial estimate

$$\hat{K}^{(t+1)} \geqslant \eta \min_{a\in\mathbb{A}} \hat{\pi}_a^{(t)} \|\boldsymbol{P}\boldsymbol{\pi}^{(t)}\|_\infty.$$

Using

$$\pi_a^{(t)} \geqslant \varepsilon - \|\hat{\boldsymbol{\pi}}^{(t)} - \boldsymbol{\pi}^*\|_1 \geqslant \varepsilon\big(1 - \sqrt{1 - \exp(-\Theta_t/\varepsilon)}\big),$$

we obtain

$$\Theta_t - \Theta_{t+1} \geqslant 4\eta^2 C_1 C_{\boldsymbol{P}}^2\varepsilon^4\big(1 - \sqrt{1 - \exp(-\Theta_t/\varepsilon)}\big)\big(1 - \exp(-\Theta_t/\varepsilon)\big).$$

Set

$$w = \sqrt{1 - \exp(-\Theta_t/\varepsilon)}, \qquad W = \sqrt{1 - \mathrm{e}^{-1}}.$$

Applying Lemma 8 and the substitution $u = \sqrt{1 - \exp(-x/\varepsilon)}$ (so $\mathrm{d}x = \frac{2u\varepsilon\,\mathrm{d}u}{1 - u^2}$) gives

$$\begin{aligned}
t - T_1 &\leqslant \ln\frac{W^2(1 - W)}{w^2(1 - w)} + \int_{\Theta_t}^{\varepsilon} \frac{\mathrm{d}x}{4\eta^2 C_1 C_{\boldsymbol{P}}^2\varepsilon^4(1 - \sqrt{1 - \exp(-x/\varepsilon)})(1 - \exp(-x/\varepsilon))} \\
&\leqslant \ln\frac{W^2(1 - W)}{w^2(1 - w)} + \frac{1}{2\eta^2 C_1 C_{\boldsymbol{P}}^2\varepsilon^3}\int_w^W \frac{\mathrm{d}u}{u(1 - u)(1 - u^2)} \\
&= \ln\frac{W^2(1 - W)}{w^2(1 - w)} + \frac{1}{2\eta^2 C_1 C_{\boldsymbol{P}}^2\varepsilon^3}\int_w^W \left(\frac{1}{u} + \frac{1 + u - u^2}{(1 - u)(1 - u^2)}\right)\mathrm{d}u
\end{aligned} \tag{5}$$

$$\leqslant 2\ln\frac{W}{w} + \frac{1}{2\eta^2 C_1 C_{\boldsymbol{P}}^2 \varepsilon^3}\left(\ln\frac{W}{w} + C_2\right),$$

where $C_2 = \int_0^W \frac{1+x-x^2}{(1-x)(1-x^2)}\mathrm{d}x$ is an absolute constant. The estimate in equation 5 used the substitution indicated above.

Finally, since

$$\Theta_t = -\varepsilon\ln(1-w^2) \leqslant -\varepsilon\ln(1-W^2)\,w^2 = \varepsilon w^2,$$

we deduce exponential decay:

$$\Theta_t \leqslant \varepsilon W \exp\Big(-(4+\eta^2 C_1^{-1}C_{\boldsymbol{P}}^{-2}\varepsilon^{-3})(t-T_1-C_2)\Big).$$

This completes the analysis: after $T_1$ iterations to reach the small region, $\Theta_t$ decreases exponentially fast with the stated rate.

# D  EXPERIMENT DETAILS AND ADDITIONAL RESULTS

This sections provides additional experiment details and results.

## D.1  GAME MATRIX CONSTRUCTION

We sample preference matrices using Algorithm 1.

---

**Algorithm 1:** Preference Matrix Sampling Algorithm

---

**Require:** Size of game $n$, rank of full-support equilibrium space $m$.
1: Sample $m$ positive vectors $\boldsymbol{v}_1, \boldsymbol{v}_2, \cdots, \boldsymbol{v}_m$ and a random $(n-m)\times(n-m)$ matrix $\boldsymbol{A}$.
2: Find an orthonormal basis $\boldsymbol{u}_1, \boldsymbol{u}_2, \cdots, \boldsymbol{u}_{n-m}$ of the orthogonal complement of the subspace $\mathrm{span}\{\boldsymbol{v}_1, \boldsymbol{v}_2, \cdots, \boldsymbol{v}_m\}$.
3: Set $\boldsymbol{M} \leftarrow [\boldsymbol{u}_1 \quad \boldsymbol{u}_2 \quad \cdots \quad \boldsymbol{u}_{n-m}]$ and $\boldsymbol{P} \leftarrow \boldsymbol{M}(\boldsymbol{A}-\boldsymbol{A}^\top)\boldsymbol{M}^\top$.
4: **return** $\frac{1}{\max_{1\leqslant i,j\leqslant n}|P_{i,j}|}\boldsymbol{P}$.

---

For $m \geqslant 1$, this guarantees that $\boldsymbol{P}\boldsymbol{v}_i = \boldsymbol{0}$, so $\frac{\boldsymbol{v}_i}{\|\boldsymbol{v}_i\|_1}$ is an equilibrium.

## D.2  CODEBASE AND BASELINES

We chose the codebase[3] open-sourced by Zhou et al. (2025) because OMD (regularized) and EGPO have been included. Necessary modifications and extensions have been made, and for completeness, we describe the implementation of baselines here. We use $\eta$ as the step size and $\beta$ as the regularization coefficient.

**OMD.**  The update is

$$\boldsymbol{\theta}^{(t+1)} = \boldsymbol{\theta}^{(t)} + \eta\boldsymbol{P}\boldsymbol{\pi}^{(t)}.$$

The implementation uses the generalized online IPO (where we choose $\beta = 1$):

$$\boldsymbol{\theta}^{(t+1)} \leftarrow \boldsymbol{\theta}^{(t)} - \frac{\eta\,|\mathbb{A}|}{4}\nabla_{\boldsymbol{\theta}}\mathcal{L}_{\mathsf{IPO}}(\boldsymbol{\theta}^{(t)};\mathsf{sg}[\boldsymbol{\pi}^{(t)}],\mathsf{Uniform},\mathsf{sg}[\boldsymbol{\pi}^{(t)}]).$$

**OMD (regularized).**  The update is shown as Equation (8) in Munos et al. (2023):

$$\boldsymbol{\pi}^{(t+1)} = \arg\max_{\boldsymbol{\pi}}\{\eta\boldsymbol{\pi}^\top\boldsymbol{P}\boldsymbol{\pi}^{(t)} - D_{\mathrm{KL}}(\boldsymbol{\pi}||\widetilde{\boldsymbol{\pi}}^{(t)})\},$$

---

[3]https://github.com/zhourunlong/EGPO

where $\widetilde{\boldsymbol{\pi}}^{(t)}(y) \propto (\boldsymbol{\pi}^{(t)}(y))^{1-\eta\beta}(\boldsymbol{\pi}_{\mathsf{ref}}(y))^{\eta\beta}$. In Appendix E.1 of Zhou et al. (2025), it is shown be to equivalent to

$$\boldsymbol{\theta}^{(t+1)} = \boldsymbol{\theta}^{(t)} - \eta\beta\left(\boldsymbol{\theta}^{(t)} - \boldsymbol{\theta}_{\mathsf{ref}} - \frac{\boldsymbol{P}\boldsymbol{\pi}^{(t)}}{\beta}\right).$$

The implementation uses the generalized online IPO:

$$\boldsymbol{\theta}^{(t+1)} \leftarrow \boldsymbol{\theta}^{(t)} - \frac{\eta\beta\,|\mathbb{A}|}{4}\nabla_{\boldsymbol{\theta}}\mathcal{L}_{\mathsf{IPO}}(\boldsymbol{\theta}^{(t)}; \boldsymbol{\pi}_{\mathsf{ref}}, \mathsf{Uniform}, \mathsf{sg}[\boldsymbol{\pi}^{(t)}]).$$

**SPPO.** The update is shown as Equation (4.6) in Wu et al. (2024):

$$\boldsymbol{\pi}^{(t+1)} = \arg\min_{\boldsymbol{\pi}} \mathbb{E}_{y\sim\boldsymbol{\pi}^{(t)}}\left(\log\frac{\boldsymbol{\pi}(y)}{\boldsymbol{\pi}^{(t)}(y)} - \eta\left(\mathcal{P}(y > \boldsymbol{\pi}^{(t)}) - \frac{1}{2}\right)\right)^2.$$

Nested-optimization is used to update the parameters: for each iteration of $t$, we apply gradient descent using a learning rate $\eta_{\mathsf{inner}}$ independent of $\eta$ for at most 10 steps, or until successive objects deviates by at most $10^{-5}$.

**MPO.** The update is shown as Equation (8) in Wang et al. (2024):

$$\boldsymbol{\pi}^{(t+1)} = \arg\max_{\boldsymbol{\pi}} \mathbb{E}_{y_1\sim\boldsymbol{\pi}, y_2\sim\boldsymbol{\pi}_{\mathsf{ref}}}\left[\mathcal{P}(y_1 > y_2) - \beta D_{\mathsf{KL}}(\boldsymbol{\pi}||\boldsymbol{\pi}_{\mathsf{ref}}) - \frac{1}{\eta}D_{\mathsf{KL}}(\boldsymbol{\pi}||\boldsymbol{\pi}^{(t)})\right].$$

As detailed in Algorithm 1 in Wang et al. (2024), each iteration of $t$ is solved with nested-optimization similar as in SPPO. Algorithm 1 in Wang et al. (2024) incorporates two tricks: (1) step size annealing: $\eta_t = \max\{1 - t/T, 5 \times 10^{-4}\}$ (where $t$ starts from 0); (2) reference policy refreshing: with every $\tau = 1000$ (our choice) steps, update $\boldsymbol{\pi}_{\mathsf{ref}} \leftarrow \boldsymbol{\pi}^{(i\tau)}$.

**ONPO.** The update is shown in Section 4.2 of Zhang et al. (2025):

$$\boldsymbol{\pi}^{(t)} = \arg\max_{\boldsymbol{\pi}}\left\{\eta\boldsymbol{\pi}^{\top}\mathcal{P}\boldsymbol{\pi}^{(t-1)} - D_{\mathsf{KL}}(\boldsymbol{\pi}||\hat{\boldsymbol{\pi}}^{(t)})\right\},$$

$$\hat{\boldsymbol{\pi}}^{(t+1)} = \arg\max_{\boldsymbol{\pi}}\left\{\eta\boldsymbol{\pi}^{\top}\mathcal{P}\boldsymbol{\pi}^{(t)} - D_{\mathsf{KL}}(\boldsymbol{\pi}||\hat{\boldsymbol{\pi}}^{(t)})\right\}.$$

Each iteration of $t$ is solved with nested-optimization twice, similar as in SPPO.

**EGPO.** The update is shown in Equations (2) and (3) in Zhou et al. (2025):

$$\boldsymbol{\theta}^{(t+1/2)} = (1 - \eta\beta)\boldsymbol{\theta}^{(t)} + \eta\beta\left(\boldsymbol{\theta}_{\mathsf{ref}} + \frac{\boldsymbol{P}\boldsymbol{\pi}^{(t)}}{\beta}\right),$$

$$\boldsymbol{\theta}^{(t+1)} = (1 - \eta\beta)\boldsymbol{\theta}^{(t)} + \eta\beta\left(\boldsymbol{\theta}_{\mathsf{ref}} + \frac{\boldsymbol{P}\boldsymbol{\pi}^{(t+1/2)}}{\beta}\right).$$

The implementation uses the generalized online IPO:

$$\boldsymbol{\theta}^{(t+1/2)} = \boldsymbol{\theta}^{(t)} - \frac{\eta\beta\,|\mathbb{A}|}{4}\nabla_{\boldsymbol{\theta}}\mathcal{L}_{\mathsf{IPO}}(\boldsymbol{\theta}^{(t)}; \boldsymbol{\pi}_{\mathsf{ref}}, \mathsf{Uniform}, \mathsf{sg}[\boldsymbol{\pi}^{(t)}]),$$

$$\boldsymbol{\theta}^{(t+1)} = \boldsymbol{\theta}^{(t)} - \frac{\eta\beta\,|\mathbb{A}|}{4}\nabla_{\boldsymbol{\theta}}\mathcal{L}_{\mathsf{IPO}}(\boldsymbol{\theta}^{(t)}; \boldsymbol{\pi}_{\mathsf{ref}}, \mathsf{Uniform}, \boldsymbol{\pi}^{(t+1/2)}).$$

## D.3 HYPERPARAMETERS

**Neural network architecture.** We use a 3-layer MLP with ReLU activation as the neural policy. The hidden dimension $d$ is set to be 10. Since we consider multi-armed bandit environments, there is no input to this policy. Hence, we use a random Gaussian noise $\mathcal{N}(0, I_d)$ as input.

**Reference policy.** For tabular policies, we initialize all the parameters to be 0 to assign the reference policy with the uniform policy. For neural policies, we use Xavier normal initialization (Glorot & Bengio, 2010) in all the middle layers, and zero initialization in the output layer, also assigning the reference policy with the uniform policy while avoiding symmetry weights and homogeneous gradients.

**Training arguments.**    For each algorithm under each scenario (tabular v.s. neural), we performed a grid search over all hyperparameters $\mathcal{V}_\eta \times \mathcal{V}_{\eta_{\text{inner}}}$:

$$\mathcal{V}_\eta = \{i \times 10^j \ : \ 1 \leqslant i \leqslant 10, -4 \leqslant j \leqslant 1\},$$
$$\mathcal{V}_{\eta_{\text{inner}}} = \{0.00003, 0.0001, 0.0003, 0.001, 0.003, 0.01, 0.03, 0.1\}.$$

For `MPO`, since $\eta_t$ is defined, we did not search over $\mathcal{V}_\eta$; for `OMD`, `OMD` (regularized), `EGPO` and `OMWU`, since no nested-optimization is needed, we did not search over $\mathcal{V}_{\eta_{\text{inner}}}$.

We use Algorithm 1 to sample 100 game matrices, and select the hyperparameters by first ensuring convergence in the most games, then prioritizing the minimal final duality gap. The final chosen hyperparameters are listed in Table 3.

Table 3: Hyperparameters for different algorithms across tabular and neural settings.

| Algorithm | Tabular LR | Tabular $\eta$ | Neural LR | Neural $\eta$ |
|:---:|:---:|:---:|:---:|:---:|
| OMD | 0.4 | | 10 | |
| OMD (regularized) | 0.001 | | 0.0002 | |
| SPPO | 0.03 | 0.1 | 0.03 | 0.1 |
| MPO | 0.0003 | | 0.09 | |
| ONPO | 0.01 | 0.01 | 0.01 | 0.01 |
| EGPO | 0.01 | | 0.09 | |
| OMWU | 9 | | 100 | |

### D.4    FULL RESULTS

Here we present full experiment results. In all the mentioned figures, duality gap values are cut off below $10^{-6}$ due to floating point precision.

**Last-iterate v.s. average-iterate convergence.**    In Figure 3, we display the duality gap of `OMD` (regularized) on tabular policies. These results illustrate the importance of last-iterate convergence guarantees.

**Comparisons between algorithms.**    Figures 4 and 5 are results of different algorithms on both tabular and neural policies. For algorithms with only average-iterate convergence guarantees, we only display the duality gap curves of their average policies. Even with extensive hyperparameter search, `SPPO`, `MPO`, and `ONPO` all fail due to slow and unstable nested-optimization.

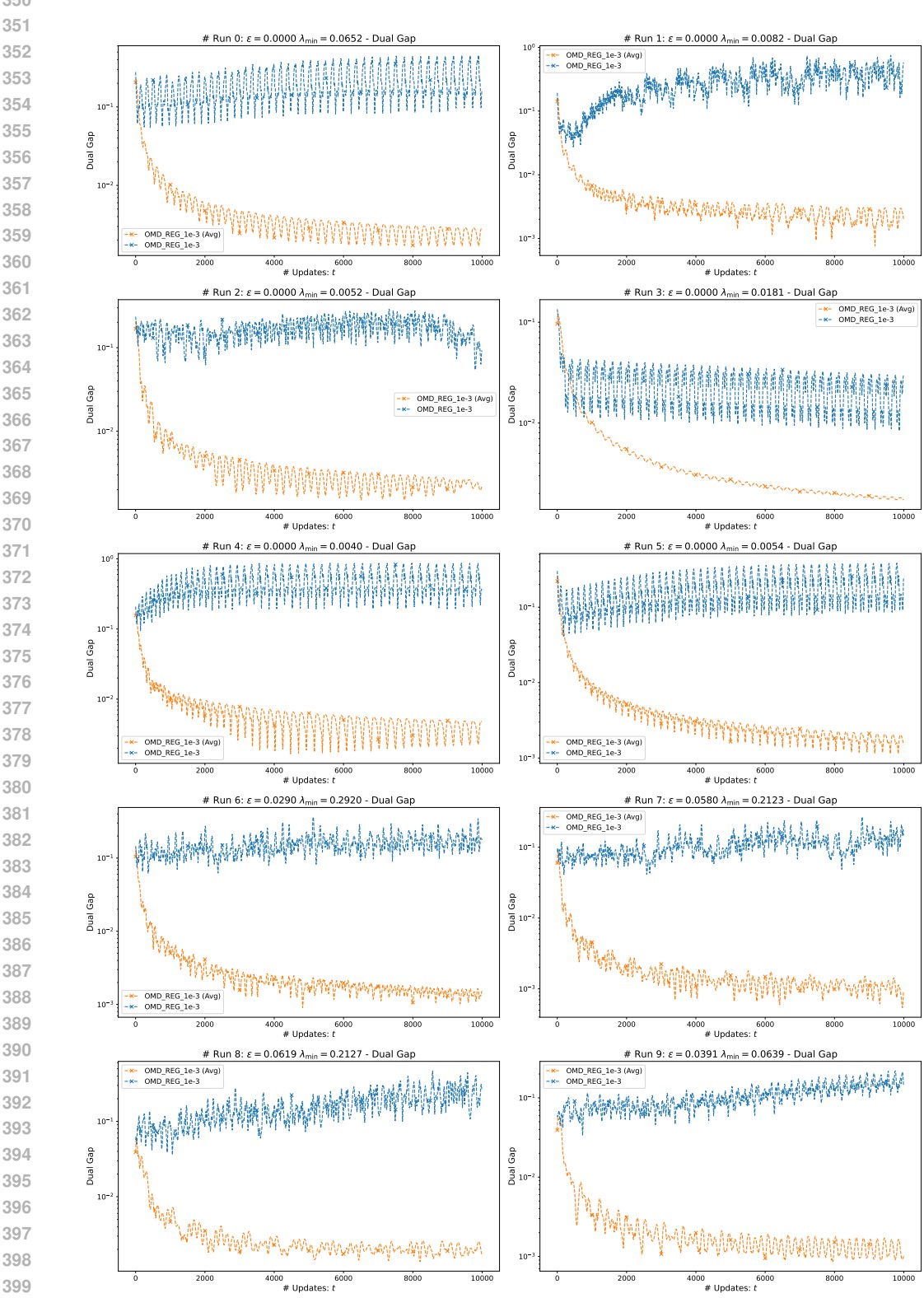

Figure 3: Duality gaps of `OMD` (regularized) applied to **tabular** policies, when evaluating the last-iterate policy $\boldsymbol{\pi}^{(t)}$ and the average-iterate policy $\frac{1}{t}\sum_{i=1}^{t}\boldsymbol{\pi}^{(i)}$.

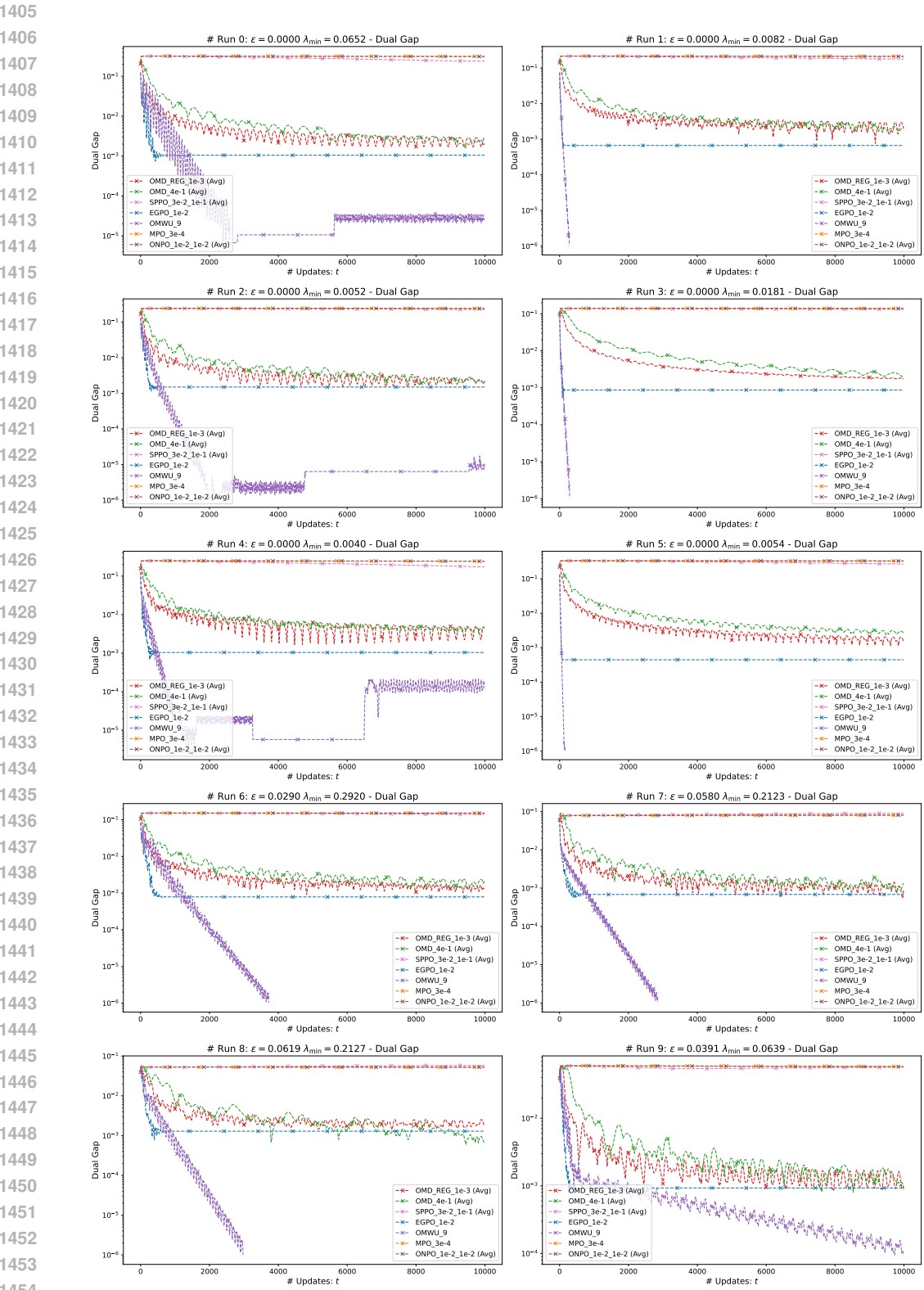

Figure 4: Duality gaps of different algorithms applied to **tabular** policies.

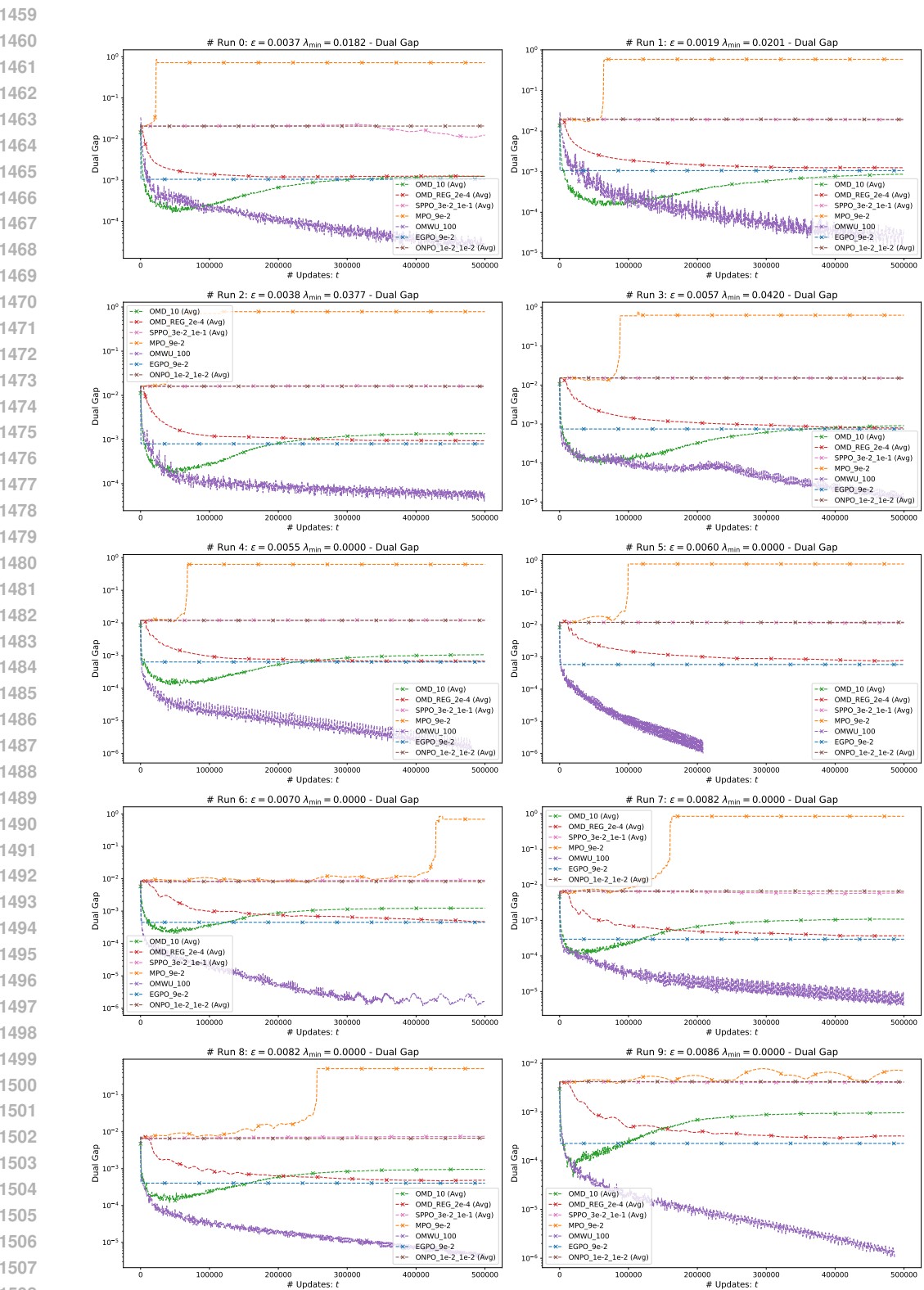

Figure 5: Duality gaps of different algorithms applied to **neural** policies.

