# OpenReview forum: "Unregularized Linear Convergence in Zero-Sum Game from Preference Feedback"
_ICLR.cc/2026/Conference — Submitted to ICLR 2026_

### Official Review · Reviewer_9ji3 · 2025-10-21

**Soundness:** 3
**Presentation:** 1
**Contribution:** 3
**Rating:** 6
**Confidence:** 4

**Summary:**

This paper investigates the Optimistic Multiplicative Weights Update (OMWU) algorithm within the Nash Learning from Human Feedback (NLHF) framework, which is used for aligning large language models. The authors provide a new theoretical analysis demonstrating that OMWU achieves last-iterate linear convergence. This result is significant because it holds under the assumption that a full-support Nash Equilibrium exists, which is a weaker and more practical condition than the uniqueness assumption required by prior work. Furthermore, their analysis improves the dependency of the convergence rate and burn-in time on problem-specific constants from exponential to polynomial. The theoretical findings are supported by experiments on both tabular and neural policy settings, where OMWU is shown to outperform several other NLHF algorithms.

**Strengths:**

- The paper's main strength is its theoretical contribution. Relaxing the uniqueness assumption to the existence of a full-support equilibrium makes the analysis applicable to a much broader class of games. The improvement from an exponential to a polynomial dependence on instance-dependent constants is a substantial leap forward, making the algorithm appear much more practical.

- The work is well-motivated by the real-world requirements of NLHF. The authors correctly identify that last-iterate convergence (as opposed to average-iterate) and computationally efficient updates (avoiding nested optimization) are crucial for deployment. By analyzing OMWU, which satisfies both of these criteria, the paper addresses key limitations of other methods in the field.

- The experiments convincingly demonstrate the effectiveness of OMWU. The algorithm is benchmarked against a comprehensive set of recent baselines, and the results in Figure 2 clearly show its superior convergence speed and accuracy. The inclusion of Figure 3, which contrasts last-iterate and average-iterate convergence for OMD, provides a powerful visual argument for the importance of the former.

- The development of a new analytical framework to characterize how OMWU escapes from sub-optimal regions of the strategy space appears to be a novel contribution that could be of broader interest to the optimization and game theory communities.

**Weaknesses:**

- The theoretical exposition, particularly the proof sketch in Section 4.2, is very dense. The heavy notation and rapid introduction of complex equations make it difficult to build an intuitive understanding of the core mechanisms behind the improved results. The paper would be more accessible if it dedicated more space to a higher-level, intuitive explanation of the key steps.

- While Assumption 1 is indeed a relaxation of uniqueness, it may still be quite strong in the context of LLMs, where the action space of possible responses is enormous. The paper would benefit from a discussion on the practical likelihood of this assumption holding in real-world preference-learning scenarios and the potential behavior of OMWU when the assumption is violated.

- The authors honestly state in the conclusion that the derived bounds for burn-in and convergence might not be tight. This is a minor point, as the polynomial dependency is the main takeaway, but it does mean the precise performance guarantees remain somewhat open.

**Questions:**

1.  Regarding Assumption 1 (existence of a full-support NE): Can you provide some intuition on what this implies for a real-world LLM alignment task? Does this mean that an optimal policy must assign a non-zero probability to every conceivable response to a prompt? If so, under what conditions might we expect this to hold for preference distributions collected from humans?

2.  In the experiments, the methods requiring nested optimization performed quite poorly. You attribute this potentially to an insufficient number of inner optimization steps (10). While I understand the practical need to limit computation, it raises the question of whether the comparison was entirely fair. Do you have any results or intuition on whether these methods would eventually converge, albeit much more slowly, if given a significantly larger number of inner-loop steps?

3.  The abstract refers to a "novel marginal convergence behavior" that is key to your analysis. Could you please elaborate on this concept in more intuitive terms? How does this behavior differ from what was understood from prior analyses, and how does it specifically allow you to avoid the exponential dependencies found in the work of Wei et al. (2020)?

---

> ### Author Response · Authors · 2025-12-02
>
> Thanks for expressing your concerns with our paper. We will first make a more intuitive explanation about the proof.
>
> At the beginning, we show the monotonicity of a KL-variance term $\Theta_t$ strictly decreases, which is previously observed (e.g., Wei et al., 2020). Next, we present Figure 1, a graph showing how $\Theta_t-\Theta_{t+1}$ evolves and breaks it into a burn-in stage and a convergence stage. Moreover, the burn-in stage is divided into a subgame case and a marginal case. To make things about how these two cases are separated, you may consider a current policy taking large probabilities on subset $\mathbb B\subset\mathbb A$, and small probabilities on $\mathbb A\backslash\mathbb B$. Moreover, we consider the subgame restricted to action set $\mathbb B$ (the exact definition of the projection between policies does not matter as the majority probability is distributed on $\mathbb B$).
>
> Subgame case happens when the projected policy is far from an equilibrium of the subgame, and in this case, an update on the full game mimics an update on the subgame (and hence the name of the case). In contrast, Marginal case happens when the projected policy is close to an equilibrium of the subgame. In this case, the log-probability of some action in $\mathbb A\backslash\mathbb B$ increases (the name of the case is from that the increase in probability, and in the KL-magnitude $\Theta_t$, are small, i.e., marginal).
>
> Since we separate these two cases, the analysis of the subgame case does not suffer from the existence of an action with exponentially small probability (which is what led to texponential dependence in previous analyses). Apart from that, the analysis of the marginal case directly shows how OMWU explores new actions. We believe our explanation will answer your question 3.
>
> Now we answer you question 2. While adding more inner optimization steps will make those methods converge eventually, we do not deem it worth the extra computation cost in reaility. We believe that our comparison is totally fair as those methods requiring nested optimization already take 3 times more time to execute than other ones with only 10 inter steps.

---

### Official Review · Reviewer_HGFL · 2025-10-30

**Soundness:** 3
**Presentation:** 3
**Contribution:** 3
**Rating:** 6
**Confidence:** 4

**Summary:**

This paper proves that OMWU achieves last-iterate linear convergence to the unregularized NE without the need for bias-inducing regularization. This result is established under a milder assumption than previous work (requiring only an existing full-support NE, not a unique one) and critically improves the convergence bounds from exponential to polynomial dependence on problem-specific constants. The analysis highlights OMWU's practical advantages for LLM alignment, as it is computationally efficient (no nested optimization) and its last-iterate guarantee is suitable for deployment. Experiments on synthetic games validate these theoretical claims, showing OMWU's superior and rapid convergence over existing regularized or average-iterate methods.

**Strengths:**

1. This paper is the first to provide a rigorous last-iterate linear convergence proof (converging to the original NE) for an unregularized algorithm (OMWU) in NLHF, thereby addressing the inherent bias introduced by regularization.

2. This paper relaxes the strict unique NE assumption required by Wei et al. (2020) to a more realistic existence of a full-support NE (Assumption 1). This relaxation enables the theory to be directly applicable to NLHF.

3. The convergence bound is improved from an exponential dependence on problem parameters to a polynomial dependence.

**Weaknesses:**

1. Although the title and introduction position the work as addressing the NLHF problem for LLM alignment, all experiments are conducted on synthetic tabular games ($n=10$) or small MLP-based policies ($n=100$). While the theoretical contribution is elegant,  It remains unclear whether OMWU can retain its theoretical advantages in realistic, high-dimensional, and non-stationary LLM fine-tuning settings.

2. The entire theoretical foundation of the paper relies on the existence of a full-support Nash equilibrium (Assumption 1). In real-world LLM alignment settings with thousands of possible responses (actions), this assumption is highly questionable. An optimal policy is likely to assign nonzero probability only to a small subset of high-quality responses, while assigning strictly zero probability to the majority.

**Questions:**

See weaknesses

---

### Official Review · Reviewer_EPUq · 2025-10-31

**Soundness:** 3
**Presentation:** 2
**Contribution:** 2
**Rating:** 4
**Confidence:** 3

**Summary:**

This paper studies alignment of large language models via Nash Learning from Human Feedback (NLHF), a zero-sum game framework that accommodates non-transitive human preferences. The authors focus on the Optimistic Multiplicative Weights Update (OMWU) algorithm and provide the first guarantee of last-iterate linear convergence under the assumption that a full-support Nash equilibrium exists. Unlike prior work, this result removes the uniqueness requirement and achieves convergence with only polynomial dependence on instance-specific constants. A novel analysis of marginal action dynamics underpins this improvement. Simulated experiments with tabular and neural policy classes validate the theory and demonstrate OMWU’s advantages over baseline NLHF algorithms.

**Strengths:**

1.The analysis relaxes prior assumptions (from unique equilibrium to a more general full-support equilibrium) and introduces a novel framework for understanding how the algorithm "escapes" from suboptimal actions, leading to faster convergence.
2.The paper is well-structured and clearly motivates the need for last-iterate convergence. It effectively contrasts its contributions with prior work, making its advancements easy to understand.
3.Experiments on synthetic data (tabular and neural policies) confirm the theoretical predictions, showing that OMWU performs strongly and achieves the expected linear convergence.

**Weaknesses:**

1. The core theoretical guarantee relies on the assumption that a Nash Equilibrium exists where every action has a non-zero probability. This may not hold in many real-world scenarios where some actions are always suboptimal, thus narrowing the theory's applicability.
2. The paper claims relevance to LLM alignment but provides no experiments on actual language models or real preference data. This makes it unclear how the theoretical gains translate to practice, leaving a gap between theory and application.

**Questions:**

none

---

### Official Review · Reviewer_WSo4 · 2025-10-31

**Soundness:** 3
**Presentation:** 2
**Contribution:** 2
**Rating:** 4
**Confidence:** 4

**Summary:**

This paper presents a new theoretical analysis of the Optimistic Multiplicative Weights Update (OMWU) algorithm for two-player zero-sum games. Assuming the Nash equilibrium has full support, the authors show that OMWU achieves a linear convergence rate after a burn-in phase. This analysis also relaxes the unique Nash equilibrium assumption made in previous works. Experiments on synthetic data further demonstrate the effectiveness of OMWU.

**Strengths:**

The theoretical results are interesting, as prior analyses of OMWU typically require the Nash equilibrium to be unique, whereas this paper relaxes that assumption by only requiring the equilibrium to have full support.

**Weaknesses:**

The presentation and organization of the paper require significant improvement. From Section 3.2 to Section 3.4, the authors attempt to connect their problem to the NLHF setting in the context of LLM alignment. However, this connection is incorrect, as the current paper studies a pure matrix game, whereas the NLHF literature (e.g., Munos et al., 2023) considers games with KL regularization terms, and it is precisely the KL regularization that makes those games relevant to the LLM alignment setting. Consequently, Section 3.2 needs to be revised, and Section 3.4 appears particularly misplaced, as the paper does not analyze the objective defined there, making the discussion disconnected and potentially misleading to readers.

Meanwhile, the title “Zero-sum Game from Preference Feedback” is also misleading, as the main algorithm and analysis focus on matrix games with full feedback. The authors do not provide any linear convergence analysis under the preference feedback setting. Overall, the paper should clearly state that the theoretical analysis applies only to the pure matrix game scenario.

The experimental results are also based on synthetic matrix games, which do not appear to have a meaningful connection to LLM alignment or RLHF scenarios.

**Questions:**

See the weaknesses part.

---

> ### Author Response · Authors · 2025-12-02
>
> Thanks for expressing your concerns with our paper. However, most of them seem to be your misunderstandings of related works. There are several works dealing with unregularized version of NLHF, such as SPPO (Wu et al., 2024) and ONPO (Zhang et al., 2025), so removing the regularization term does not invalidate the algorithm. Also, previous convergence analysis are typically done on the tabular case for simplicity. You make fine how to adapt the algorithm for LLM fine-tuning in Section 5.2, and the EGPO paper (Zhou et al., 2025) is also a good reference.

---

### Author Response · Authors · 2025-12-02

I would like to address the commonly mentioned questions here. The first one is about the justification of Assumption 1 in practical settings. Intuitively speaking, if the current policy takes small probabilities on suboptimal actions, then an update on the full game would be approximate to an update on the subgame where only optimal actions are allowed, and in this case, the convergence result is already proved. In addition, the monotonicity of $\Theta_t$ does not require Assumption 1. As a result, we believe this assumption can be loosened. Also, we conducted experiments for games without full-support equilibriums, and these cases are identified by $\varepsilon=0$ (generally, $\varepsilon=\min_{a\in\mathbb A}\pi_a$ and $\pi$ is an equilibrium policy minimizing its negative entropy). Moreover, some bad actions are easy to eliminate in practical scenarios.

The second question is about the lack of experiments. We admit that we currently cannot finish LLM fine-tuning experiments due to resource constraints. However, we believe in the success of OMWU, as the EGPO paper (Zhou et al., 2025) exhibits promising results when a tabular algorithm is adapted for LLM fine-tuning.

---

### Meta-Review · Area_Chair_C2uf · 2026-01-05

**Summary:**

This paper provides the first last-iterate linear convergence guarantee for  Optimistic Multiplicative Weights Update (OMWU) in unregularized Nash Learning from Human Feedback, relaxing the unique NE assumption from prior work to only requiring a full-support NE. Reviewers generally appreciated the theoretical contribution but raised consistent concerns about: (1) the practical relevance of the full-support assumption in LLM settings with large action spaces, (2) the absence of LLM experiments despite the paper's framing around alignment, and (3) presentation issues including potentially misleading title/positioning.

**Reviewer Concerns:**

- Authors provided intuition for why Assumption 1 may be loosened in practice and noted experiments identifying cases without full-support equilibria
- Authors clarified the proof structure in response to Reviewer 9ji3's questions about marginal convergence behavior


- No LLM experiments remain a gap (authors cite resource constraints but this weakens the claimed relevance to alignment)
- The disconnect between the paper's NLHF/LLM framing and its actual matrix game analysis (Reviewer WSo4's core concern) was not fully resolved
- Full-support assumption remains questionable for realistic LLM scenarios

**Reviewer Scores:**

- Reviewer WSo4 (4): Likely unchanged; author response was somewhat dismissive of valid concerns about framing
- Reviewer EPUq (4): Likely unchanged; main concerns not addressed
- Reviewer HGFL (6): Likely unchanged; concerns acknowledged but not resolved
- Reviewer 9ji3 (6): Possibly slight increase given helpful clarifications on proof intuition

---

### Decision · Program_Chairs · 2026-01-26

Reject